# Exploring high-quality microbial genomes by assembling short-reads with long-range connectivity

Zhenmiao Zhang [1,8], Jin Xiao[1,8], Hongbo Wang[1], Chao Yang[1], Yufen Huang[2], Zhen Yue [3], Yang Chen[4], Lijuan Han[5], Kejing Yin[1,6], Aiping Lyu [7], Xiaodong Fang[2,3,5] & Lu Zhang [1,6] ✉

Although long-read sequencing enables the generation of complete genomes for unculturable microbes, its high cost limits the widespread adoption of long-read sequencing in large-scale metagenomic studies. An alternative method is to assemble short-reads with long-range connectivity, which can be a cost-effective way to generate high-quality microbial genomes. Here, we develop Pangaea, a bioinformatic approach designed to enhance metagenome assembly using short-reads with long-range connectivity. Pangaea leverages connectivity derived from physical barcodes of linked-reads or virtual barcodes by aligning short-reads to long-reads. Pangaea utilizes a deep learning-based read binning algorithm to assemble co-barcoded reads exhibiting similar sequence contexts and abundances, thereby improving the assembly of high- and medium-abundance microbial genomes. Pangaea also leverages a multi-thresholding algorithm strategy to refine assembly for low-abundance microbes. We benchmark Pangaea on linked-reads and a combination of short- and long-reads from simulation data, mock communities and human gut metagenomes. Pangaea achieves significantly higher contig continuity as well as more near-complete metagenome-assembled genomes (NCMAGs) than the existing assemblers. Pangaea also generates three complete and circular NCMAGs on the human gut microbiomes.

Metagenome assembly is one of the main steps to reconstruct microbial genomes from culture-free metagenomic sequencing data[1]. Cost-effective short-read sequencing technologies have been widely applied to generate high-quality microbial reference genomes from large cohorts of human gut microbiomes[2–4]. However, the short-read length (100–300bps) may not allow us to resolve intra-species repetitive regions and inter-species conserved regions[5] or to achieve complete microbial genomes. The emerging long-read sequencing technologies such as PacBio continuous long-read sequencing (PacBio CLR)[6], Oxford Nanopore sequencing (ONT)[7] and PacBio HiFi sequencing[8], have shown their superiority to short-read sequencing in generating metagenome-assembled genomes (MAGs) with high continuity or producing complete and circular MAGs using long-range connectivity they provided[9–11]. Despite potential benefits, the high cost of long-read sequencing makes deep sequencing impracticable and continues to hinder its application in population-scale or clinical

[1]Department of Computer Science, Hong Kong Baptist University, Hong Kong, China. [2]BGI Research, Shenzhen 518083, China. [3]BGI Research, Sanya 572025, China. [4]State Key Laboratory of Dampness Syndrome of Chinese Medicine, The Second Affiliated Hospital of Guangzhou University of Chinese, Guangzhou, China. [5]Department of Scientific Research, Kangmeihuada GeneTech Co., Ltd (KMHD), Shenzhen, China. [6]Institute for Research and Continuing Education, Hong Kong Baptist University, Shenzhen, China. [7]School of Chinese Medicine, Hong Kong Baptist University, Hong Kong, China. [8]These authors contributed equally: Zhenmiao Zhang, Jin Xiao. ✉e-mail: ericluzhang@hkbu.edu.hk

studies[12]. In our previous study[13], we observed that long-reads generated fewer high-quality MAGs than short-reads due to insufficient sequencing depth. As an alternative way to deep long-read sequencing, some studies[13–16] suggested utilizing cost-effective short-reads with long-range connectivity for metagenome assembly. The long-range connectivity could be derived from physical barcodes (e.g., linked-reads) or virtual links by long-fragment sequencing technologies (e.g., long-reads).

Linked-read sequencing attaches identical barcodes (physical barcodes) to the short-reads if they are derived from the same long DNA fragment. Before the discontinuation, 10x Chromium was the most widely used linked-read sequencing technology, generating contigs with high continuity and producing more near-complete metagenome-assembled genomes (NCMAGs; Methods) than short-read sequencing[13,14]. However, co-barcoded short-reads of 10x Genomics have a high chance of being derived from multiple DNA fragments (the average number of fragments per barcode [$N_{F/B}$] is 16.61; Methods and Supplementary Note 1), which may complicate the deconvolution of complex microbial communities. Recently, MGI and Universal Sequencing Technology released their linked-read sequencing technologies, namely single-tube Long Fragment Read (stLFR)[17] and Transposase Enzyme-Linked Long-read Sequencing (TELL-Seq)[18]. The barcoding reactions of these technologies occur on billions of microbeads, leading to much higher barcode specificity ($N_{F/B} = 1.54$ for stLFR; $N_{F/B} = 4.26$ for TELL-Seq; Supplementary Note 1). The co-barcoded short-reads of these technologies are more likely to come from the same genomic regions.

Several tools have been developed for linked-read assembly: (i) Athena[14] fills the gaps between contigs by recruiting the 10x Genomics co-barcoded reads for local assembly; (ii) cloudSPAdes[19] reconstructs the long DNA fragments in the assembly graph for solving the shortest superstring problem to improve contig continuity; (iii) Supernova[20] was developed for human genome diploid assembly by allowing two paths in megabubble structures based on a series of modification of assembly graph using 10x co-barcoded reads; (iv) MetaTrass[21] groups stLFR co-barcoded reads by reference-based taxonomic annotation and applies Supernova to assemble the genome of each identified species. With the exception of MetaTrass, all the other three tools were developed for 10x Genomics linked-reads with low barcode specificity. We excluded MetaTrass for comparison because it relies on the available microbial reference genomes and thus has a limited capability to discover novel species. There is a lack of an efficient tool that could fully exploit the long-range connectivity of short-reads from barcodes with high specificity to improve de novo metagenome assembly.

Long-range connectivity could also be provided by the other long-fragment sequencing technologies (e.g., long-reads), which can be used in conjunction with short-reads for hybrid assembly. The hybrid assembly is typically performed by combining deep short-read sequencing with shallow long-read sequencing. It takes advantage of the high base quality of short-reads for contig assembly and the long-range connectivity from long-reads for contig extension. Several hybrid assemblers were developed for metagenome assembly, for example (i) hybridSPAdes[16] maps long-reads to the assembly graph from short-reads and utilizes the long-range connectivity to resolve uneven path depth and repetitive sequences in the graph; (ii) OPERA-MS[15] aligns long-reads to the contigs assembled from short-reads to construct a scaffold graph and groups contigs based on microbial reference genomes followed by gap filling in each cluster; and (iii) MetaPlatanus[22] extends contigs from the short-read assembly using long-range connectivity from long-reads, species-specific sequence compositions, and read depth.

Previous studies showed that read subsampling was an effective strategy for assembling large complex metagenomic datasets[23]. It could improve the assembly of high-abundance microbes[24], but result

in poor quality in assembling low-abundance microbes[25] due to insufficient reads. The read binning strategy has been proven advantageous in metagenome assembly[26–28]. It could be a more sophisticated read subsampling strategy to improve the assemblies of high- and medium-abundance microbes. However, the existing tools are impractical when it comes to handling millions of short-reads within acceptable time and memory limitations[26].

We introduce Pangaea to improve metagenome assembly using short-reads with long-range connectivity based on three modules. (i) Firstly, Pangaea performs co-barcoded short-read clustering to reduce the complexity of metagenomic sequencing data. Pangaea groups co-barcoded short-reads rather than grouping independent short-reads, as the co-barcoded reads are highly likely to be from the same long fragments (Supplementary Note 1). Short-reads from each cluster are assembled individually, which is believed to result in high-quality assemblies for high- and medium-abundance microbes. This is because the short-reads within the same cluster exhibit lower complexity compared to the original dataset. (ii) Secondly, Pangaea adopts a multi-thresholding reassembly step to refine the assembly of low-abundance microbes using different abundance thresholds to handle the uneven abundances of microbes (Methods). The data from high-abundance microbes are gradually removed from the assembly graph, thus the sequences from various levels of low-abundance microbes are preserved. (iii) Thirdly, Pangaea integrates the assemblies from the above two modules, original short-read assembly and local assembly (by Athena) to improve contig continuity (Methods).

We benchmarked Pangaea using short-reads with physical barcodes from linked-read simulation of mock metagenomes, and linked-read sequencing of the mock and human gut metagenomes. We evaluated its generalizability using short-reads with virtual barcodes generated from their alignments to long-reads (Methods). For linked-reads, we compared Pangaea with two short-read assemblers (metaSPAdes[29] and MEGAHIT[30]) and three linked-read assemblers (cloudSPAdes, Supernova and Athena). We found Pangaea achieved substantially better contig continuity and more NCMAGs than the other tools on all datasets. It also generated three complete and circular microbial genomes for the three real complex microbial communities. For short-reads with virtual barcodes, Pangaea could substantially improve contig continuity and generate better assemblies for both high- and low-abundance microbes than the short-read and hybrid assemblers.

## Results
### Workflow of Pangaea
Pangaea is a de novo metagenome assembler designed for short-reads with long-range connectivity represented by their attached barcodes (Fig. 1a). The barcodes can be generated physically (from linked-read sequencing) or virtually (from long-reads; Methods). The core algorithms of Pangaea are designed to reduce the complexity of metagenomic sequencing data (for high- and medium-abundance microbes) and deal with the uneven abundances of involved microbes (for low-abundance microbes) based on the barcodes with high specificity. Pangaea contains three main modules: (i) Co-barcoded read binning. This module is intended to reduce the complexity of metagenomic sequencing data and is mainly used to improve the assemblies of high- and medium-abundance microbes. Pangaea extracts $k$-mer histograms and tetra-nucleotide frequencies (TNFs; Methods) of co-barcoded reads and represents them in low-dimensional latent space by Variational Autoencoder (VAE; Methods; Supplementary Note 2). Pangaea adopts a weighted sampling strategy on training VAE to balance the number of co-barcoded short-reads from microbes with different abundances (Methods). Pangaea utilizes RPH-kmeans (k-means based on random projection hashing)[31] to group co-barcoded short-reads in the latent space[31], which is beneficial for bins with uneven sizes (Methods). Short-reads from the same bin have a high chance of

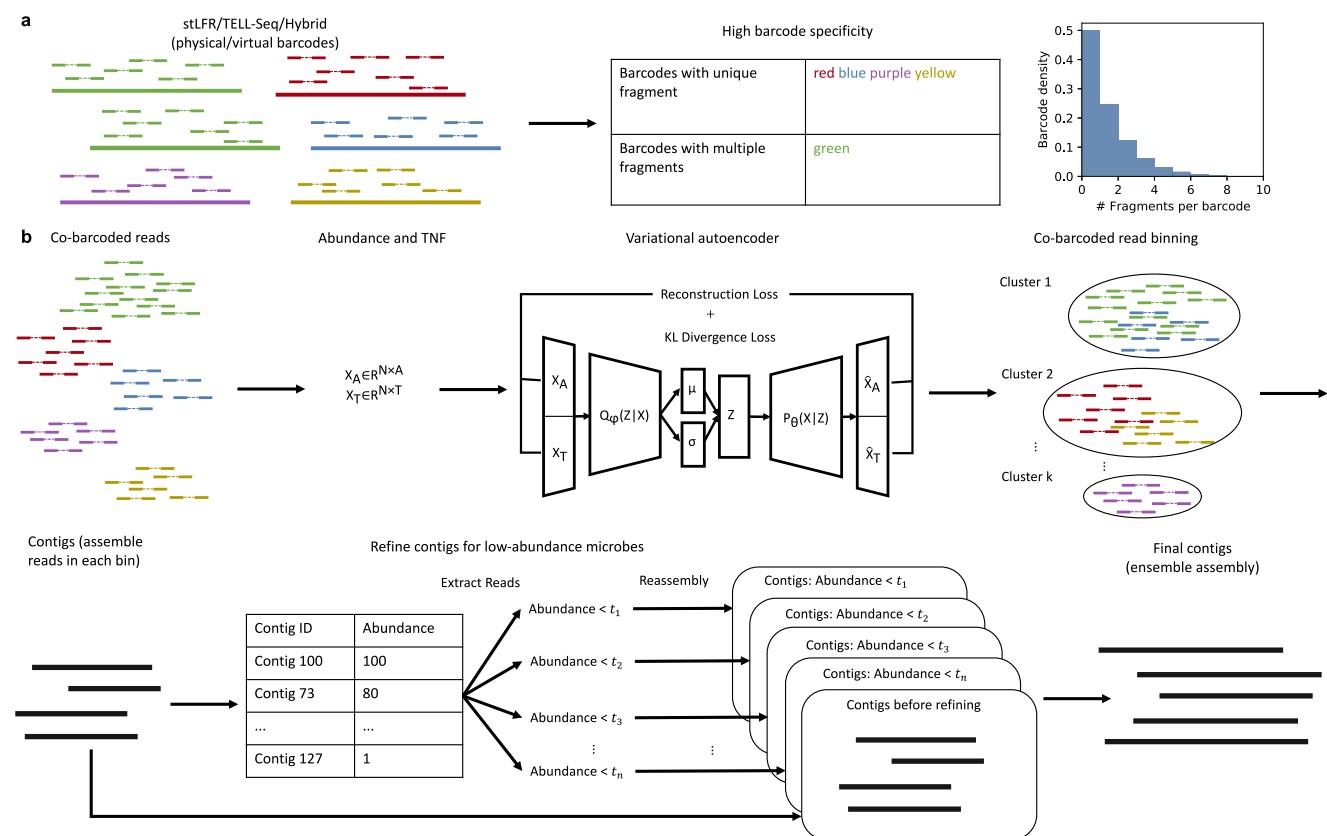

**Fig. 1 | Workflow of Pangaea. a** Pangaea could assemble reads with physical barcodes from linked-read sequencing, or virtual barcodes from aligning short-reads to long-reads. Linked-reads of stLFR and TELL-Seq are with high barcode specificity. **b** Pangaea extracts features including *k*-mer histograms and TNFs from co-barcoded reads. The features are concatenated and used to represent reads in low-dimensional latent space using a variational autoencoder. The embeddings of co-barcoded reads are clustered by RPH-kmeans. Pangaea assembles the reads from each bin independently and adopts a multi-thresholding reassembly strategy to improve the assemblies for low-abundance microbes. Ensemble assembly integrates the contigs from different strategies using OLC algorithm.

originating from the same microbe. These short-reads are then independently assembled. (ii) Multi-thresholding reassembly. Read binning may divide co-barcoded short-reads from the same low-abundance microbe into different bins. It could lead to poor assembly performance for these microbes due to insufficient data. Pangaea improves the assemblies of low-abundance microbes by collecting and reassembling the linked-reads that cannot be aligned to high-depth contigs obtained from read binning. The high-depth contigs are defined based on different depth thresholds (Fig. 1b; Methods). (iii) Ensemble assembly. This module is to eliminate the impact of mis-binning on final assembly results. Pangaea merged the assemblies from the previous two modules, the local assembly of Athena and the original short-read assembly using OLC assembly strategy (Methods). For short-reads with virtual barcodes, Pangaea would additionally integrate the contigs from the selected hybrid assembler using quickmerge (Methods).

### Benchmarking datasets

We adopted metagenomic sequencing datasets from three mock microbial communities and three fecal samples to benchmark Pangaea with the existing assemblers. The three mock communities are with known compositions and all reference genomes of the involved microbes are available: (i) ATCC-MSA-1003, containing 20 strains with different abundances varying from 0.02% to 18% (Supplementary Table 1); (ii) CAMI-high, CAMI high-complexity microbial community including 596 microbial genomes and 478 circular elements[32]; and (iii) ZYMO (ZymoBIOMICS™ Microbial Community Standard II (Log Distribution)), with extremely imbalanced abundance for 10 strains from

0.000089% to 89.1%[33]. ATCC-MSA-1003 was sequenced using stLFR (132.95 Gb), TELL-Seq (173.28 Gb) and 10x Genomics linked-reads (100.38Gb); stLFR linked-reads were simulated for CAMI-high and ZYMO (Methods); the three fecal samples (S1: 136.60 Gb, S2: 131.59 Gb, and S3: 50.74 Gb) were sequenced using stLFR linked-reads (Supplementary Table 2; Supplementary Fig. 1; Methods).

### Co-barcoded read binning improves assemblies for microbes with high and medium abundances

Co-barcoded read binning is a core step of Pangaea and it could significantly improve the assembly for high- and medium-abundance microbes, particularly for the real complex microbial communities.

We compared the metagenome assemblies of Pangaea with ($ASM_B$) and without ($ASM_{\neg B}$) read binning to investigate the impact of co-barcoded read binning on final assembly results. For ATCC-MSA-1003, $ASM_B$ had a higher overall NA50 (649.67 kb) than $ASM_{\neg B}$ (601.67 kb; Supplementary Data 1). Considering only medium and high-abundance strains, $ASM_B$ achieved higher NGA50s for 7 out of the 10 strains with abundances > 1% (Supplementary Data 1). Although there were no genomes with an abundance > 1% in CAMI-high, $ASM_B$ still had a higher overall NA50 (212.49 kb) than $ASM_{\neg B}$ (208.31 kb; Supplementary Data 1). We observed $ASM_B$ and $ASM_{\neg B}$ generated comparable N50s on the three human gut microbiomes (Supplementary Data 1).

We further evaluated the assemblies with respect to MAG qualities (completeness, contamination, and RNA annotations) after contig binning on CAMI-high and the three human gut microbiomes (Methods). For ATCC-MSA-1003, the reference genomes are available, and

the community design is simple (only 20 strains); therefore, contig binning is unnecessary. We observed $ASM_B$ generated one more NCMAG than $ASM_{\neg B}$ for CAMI-high ($ASM_B = 14$, $ASM_{\neg B} = 13$; Supplementary Data 2). For the human gut microbiomes, $ASM_B$ obtained more NCMAGs than $ASM_{\neg B}$ on all the three samples (S1: $ASM_B = 24$, $ASM_{\neg B} = 20$; S2: $ASM_B = 17$, $ASM_{\neg B} = 11$; S3: $ASM_B = 9$, $ASM_{\neg B} = 8$; Supplementary Data 2). These NCMAGs from $ASM_B$ were commonly observed from high-abundance microbes (average depths of NCMAGs: S1 = 526.6X; S2 = 211.19X; S3 = 256.52X).

Our results showed that the number of read bins ($k$) could influence both the precision and recall of read binning (a large $k$ resulted in high binning precision and low recall; Supplementary Fig. 2). The $k$ is a trade-off between generating read bins with low complexities (large $k$) or keeping more reads from the same microbes in the same bin (small $k$). $k$ for read binning was set linear to the biodiversity of a metagenomic sample ($k = a*Shannon\_Diversity$; Supplementary Table 2; Methods). To determine the coefficient $a$, we chose the $k$ ($k = 30$) that worked well on all three real metagenomic datasets and calculated the coefficient $a$ as 8 by linear regression. This setting of $k$ is applicable to both low- (ATCC-MSA-1003) and high-complexity (CAMI-high) datasets, and the assembly results seemed robust if $k$ was not shifted too much from the value calculated from the formula (15 for ATCC-MSA-1003, Supplementary Fig. 3; 37 for CAMI-high, Supplementary Data 3).

## Multi-thresholding reassembly improves assemblies for microbes with low abundance

Pangaea improves assemblies of low-abundance microbes by gradually removing the reads from high-abundance microbes from the assembly graph with multiple abundance thresholds (represented by $T$). As this module aims to improve low-abundance microbial assembly, we only consider reads from contigs with average depths lower than 10x (ultra-low) and 30x (low). The two thresholds have been validated by ATCC-MSA-1003 (Supplementary Note 3).

To demonstrate the performance of multi-thresholding reassembly, we compared the assemblies with and without multi-thresholding reassembly on both the TELL-Seq dataset of ATCC-MSA-1003 (low-complexity), and the stLFR dataset of CAMI-high (high-complexity). In the evaluation, we only consider the strains with abundances <1% as low-abundance microbes. Specifically, we identified 10 such strains in the ATCC-MSA-1003 dataset and 596 strains in the CAMI-high dataset. On ATCC-MSA-1003, we found this module could increase the NGA50s of 5 low-abundance microbes (out of 6 strains with non-zero NGA50s; Supplementary Data 4). It could generate more sequences from the long contigs (>10 kb) of 6 low-abundance microbes (out of 7 strains with contigs longer than 10 kb; Supplementary Data 4). The assembly with multi-thresholding reassembly generated significantly higher genome fractions for the 596 low-abundance microbes than the assembly without this module (Wilcoxon paired rank-sum test $p = 3.97e\text{-}05$, 95 percent confidence interval = [0.03, 0.12], effect size statistic = 0.168; Supplementary Data 5).

## Barcode specificity is critical for linked-read assembly

We applied Pangaea to linked-reads from 10x Genomics, TELL-Seq, and stLFR of ATCC-MSA-1003 to investigate the impact of barcode specificity on the performance of Pangaea (Supplementary Table 2; Methods). The linked-reads from stLFR and TELL-Seq yielded much lower $N_{F/B}$ (stLFR: 1.54, TELL-Seq: 4.26) compared to those obtained from 10x Genomics (10x Genomics: 16.61; Supplementary Note 1). The contigs from Pangaea on stLFR and TELL-Seq datasets had substantially higher N50s (1.44 times on average; Supplementary Table 3) and higher overall NA50s (1.43 times on average; Supplementary Table 3) than the assembly on 10x Genomics linked-reads. For those 15 strains with abundance > 0.1% (Supplementary Data 6), the assembly on stLFR linked-reads achieved significantly higher

strain NA50s ($p = 0.0353$; 95 percent confidence interval = [4055.0, 671843.5]; effect size statistic = 0.543; Methods) and NGA50s ($p = 0.0479$; 95 percent confidence interval = [27.5, 588585.5]; effect size statistic = 0.511; Methods) than those on 10x Genomics dataset. The same trend was also observed between the assemblies on TELL-Seq and 10x Genomics datasets (Supplementary Data 6). For the remaining 5 strains with abundances of 0.02%, all datasets cannot be assembled with high genome fractions, making it infeasible to compare their NGA50s (Supplementary Data 7). These results suggest that linked-reads with high barcode specificity could produce better metagenome assemblies using Pangaea.

## Pangaea generated high-quality metagenome assemblies on mock and simulated linked-read datasets

We benchmarked Pangaea with Athena, Supernova, cloudSPAdes, MEGAHIT, and metaSPAdes on MSA-ATCC-1003 and CAMI-high datasets with available reference genomes. For the two short-read assemblers MEGAHIT and metaSPAdes, the input were barcode-removed short-reads from linked-reads. For the other assemblers, linked-reads were used. For TELL-Seq of ATCC-MSA-1003 (Table 1; Fig. 2b), Pangaea achieved the highest N50 (1.36Mb) and overall NA50 (649.47 kb) when compared with the statistics achieved by Athena (N50: 466.50 kb; NA50: 361.57 kb), Supernova (N50: 102.76 kb; NA50: 97.31 kb), cloudSPAdes (N50: 127.42 kb; NA50: 118.16 kb), MEGAHIT (N50: 128.07 kb; NA50: 112.51 kb) and metaSPAdes (N50: 112.34 kb; NA50: 105.63 kb) (Fig. 2a, c). When considering those 15 strains with abundances >0.1% (Supplementary Data 6), Pangaea still generated significantly higher strain NA50s (Fig. 2e) and NGA50s (Fig. 2h) than Athena (NA50: $p = 8.36e\text{-}3$, 95 percent confidence interval = [2316, 384961], effect size statistic = 0.681; NGA50: $p = 8.36e\text{-}3$, 95 percent confidence interval = [3625, 415993], effect size statistic = 0.681), Supernova (NA50: $p = 3.05e\text{-}4$, 95 percent confidence interval = [237299, 1957688], effect size statistic = 0.932; NGA50: $p = 3.05e\text{-}4$, 95 percent confidence interval = [252335, 1986144], effect size statistic = 0.932), cloudSPAdes (NA50: $p = 6.10e\text{-}5$, 95 percent confidence interval = [278575, 1917580], effect size statistic = 1.035; NGA50: $p = 6.10e\text{-}5$, 95 percent confidence interval = [255292, 1910689], effect size statistic = 1.035), MEGAHIT (NA50: $p = 6.10e\text{-}5$, 95 percent confidence interval = [321189, 1890966], effect size statistic = 1.035; NGA50: $p = 6.10e\text{-}5$, 95 percent confidence interval = [277135, 1890966], effect size statistic = 1.035) and metaSPAdes (NA50: $p = 6.10e\text{-}5$, 95 percent confidence interval = [308107, 1862172], effect size statistic = 1.035; NGA50: $p = 6.10e\text{-}5$, 95 percent confidence interval = [271810.5, 1839500.0], effect size statistic = 1.035). A comparable trend was observed on the assemblies of 10x Genomics and stLFR linked-reads (Table 1; Fig. 2d, g, f, i). For the 5 strains with the lowest abundance (0.02%) of ATCC-MSA-1003, the assemblies of Pangaea had much higher genome fractions than those of Athena (9.40 times on average) and Supernova (47.87 times on average) on all three technologies (Supplementary Data 7), suggesting more genomic sequences could be assembled by Pangaea for low-abundance microbes.

For simulated linked-reads from CAMI-high, Pangaea generated the highest total assembly length, genome fraction, N50 (1.87 times on average), overall NA50 (1.61 times on average), NA50 per strain (1.44 times on average), and NGA50 per strain (1.58 times on average) than the other assemblers (Table 1). Although Pangaea and Athena got comparable overall NA50s (Pangaea produced more sequences), the NGA50 per strain of Pangaea was much higher than that of Athena (Pangaea = 54.41 kb, Athena = 49.90 kb; Table 1). Pangaea generated the largest number of genomes for which the assemblies covered at least 50% (non-zero NGA50s; Pangaea = 195, Athena = 180, Supernova = 175, cloudSPAdes = 177, metaSPAdes = 177, MEGAHIT = 180; Table 1). These results indicate Pangaea performed well on both the high-complexity dataset and the low-abundance microbes.

**Table 1 | Assembly statistics for different assemblers using the barcode-removed short-reads or linked-reads on mock communities**

| | Pangaea | Athena | Supernova | cloudSPAdes | MEGAHIT | metaSPAdes |
|---|---|---|---|---|---|---|
| **ATCC-MSA-1003 (stLFR)** | | | | | | |
| Total assembly length | **59,484,233** | 52,159,846 | 35,226,545 | - | 55,506,708 | 57,225,487 |
| Genome fraction (%) | **84.43** | 77.12 | 52.21 | - | 81.99 | 83.99 |
| Longest alignment | **2,853,278** | 2,281,647 | 1,105,108 | - | 883,580 | 883,552 |
| Overall N50 | **1,619,916** | 875,747 | 243,194 | - | 127,879 | 132,556 |
| Overall N70 | 614,609 | 615,896 | 132,825 | - | **63,957** | 63,879 |
| Overall N90 | 5,248 | **110,802** | 50,969 | - | 5,222 | 3,688 |
| Overall NA50 | **731,990** | 677,911 | 215,052 | - | 116,995 | 125,586 |
| NA50 per strain | **628,059** | 576,621 | 145,646 | - | 140,463 | 134,477 |
| NGA50 per strain | **677,353** | 575,371 | 137,023 | - | 133,877 | 133,978 |
| **ATCC-MSA-1003 (TELL-Seq)** | | | | | | |
| Total assembly length | 61,990,266 | 60,847,375 | 56,748,937 | **62,316,993** | 60,291,592 | 60,648,311 |
| Genome fraction (%) | **82.63** | 81.99 | 76.46 | 82.44 | 82.10 | 82.46 |
| Longest alignment | **4,968,123** | 4,968,084 | 1,096,372 | 884,364 | 867,473 | 776,102 |
| Overall N50 | **1,360,322** | 466,498 | 102,757 | 127,419 | 128,069 | 112,342 |
| Overall N70 | **465,633** | 184,646 | 41,121 | 56,614 | 54,222 | 49,466 |
| Overall N90 | 8,045 | **9,929** | 8,084 | 6,051 | 5,893 | 5,429 |
| Overall NA50 | **649,672** | 361,569 | 97,312 | 118,159 | 112,513 | 105,630 |
| NA50 per strain | **838,457** | 483,734 | 123,277 | 129,001 | 122,932 | 119,253 |
| NGA50 per strain | **887,107** | 485,196 | 121,657 | 129,531 | 121,938 | 118,391 |
| **ATCC-MSA-1003 (10x)** | | | | | | |
| Total assembly length | 58,860,253 | 52,292,807 | **89,828,047** | - | 56,558,134 | - |
| Genome fraction (%) | **83.23** | 77.19 | 75.08 | - | 82.74 | - |
| Largest alignment | 2,277,835 | **2,278,264** | 974,529 | - | 883,602 | - |
| Overall N50 | **1,033,793** | 601,544 | 32,128 | - | 151,002 | - |
| Overall N70 | **564,696** | 356,490 | 12,725 | - | 73,366 | - |
| Overall N90 | 13,052 | **73,456** | 4,075 | - | 6,715 | - |
| Overall NA50 | **483,416** | 453,155 | 30,194 | - | 132,728 | - |
| NA50 per strain | 421,157 | 328,424 | 93,097 | - | 141,574 | - |
| NGA50 per strain | 441,029 | 334,491 | 89,993 | - | 143,179 | - |
| **CAMI-high** | | | | | | |
| Total assembly length | **799,834,811** | 773,344,531 | 757,315,614 | 752,648,273 | 772,095,404 | 759,108,499 |
| Genome fraction (%) | **27.71** | 27.33 | 26.69 | 26.66 | 27.46 | 27.01 |
| Longest alignment | **3,402,679** | 3,402,661 | 2,700,464 | 2,525,441 | 2,378,494 | 2,518,442 |
| Overall N50 | **254,141** | 211,169 | 135,915 | 117,495 | 114,277 | 133,877 |
| Overall N70 | **124,320** | 108,630 | 61,513 | 54,187 | 52,909 | 62,467 |
| Overall N90 | **38,713** | 38,425 | 15,770 | 10,227 | 10,946 | 12,521 |
| Overall NA50 | **212,494** | 201,566 | 132,676 | 113,425 | 110,658 | 132,603 |
| NA50 per strain | **66,310** | 66,060 | 46,464 | 40,507 | 39,340 | 45,216 |
| NGA50 per strain | **54,412** | 49,896 | 34,201 | 31,266 | 28,301 | 34,547 |
| Non-zero NGA50 | **195** | 180 | 175 | 177 | 180 | 177 |
| **ZYMO** | | | | | | |
| Total assembly length | **36,094,665** | 35,254,802 | 25,103,660 | 35,843,842 | 35,358,538 | 35,511,020 |
| Genome fraction (%) | **49.24** | 48.14 | 34.27 | 47.99 | 48.38 | 48.46 |
| Longest alignment | **2,839,942** | 2,717,703 | 768,486 | 1,012,282 | 1,079,942 | 847,644 |
| Overall N50 | **1,094,665** | 761,749 | 288,912 | 210,427 | 124,248 | 191,688 |
| Overall N70 | **638,009** | 445,656 | 139,701 | 136,651 | 70,860 | 106,796 |
| Overall N90 | **180,146** | 157,620 | 45,025 | 60,073 | 25,980 | 44,425 |
| Overall NA50 | **1,072,622** | 760,284 | 226,947 | 208,894 | 116,738 | 191,688 |
| NA50 per strain | **1,066,276** | 927,278 | 136,210 | 245,562 | 179,245 | 183,935 |
| NGA50 per strain | **1,083,616** | 938,270 | 132,015 | 246,262 | 177,318 | 179,371 |

The missing values for cloudSPAdes and metaSPAdes were because they required over 2TB memory on the relevant datasets that exceeded our server limit.
The highest values are in bold.

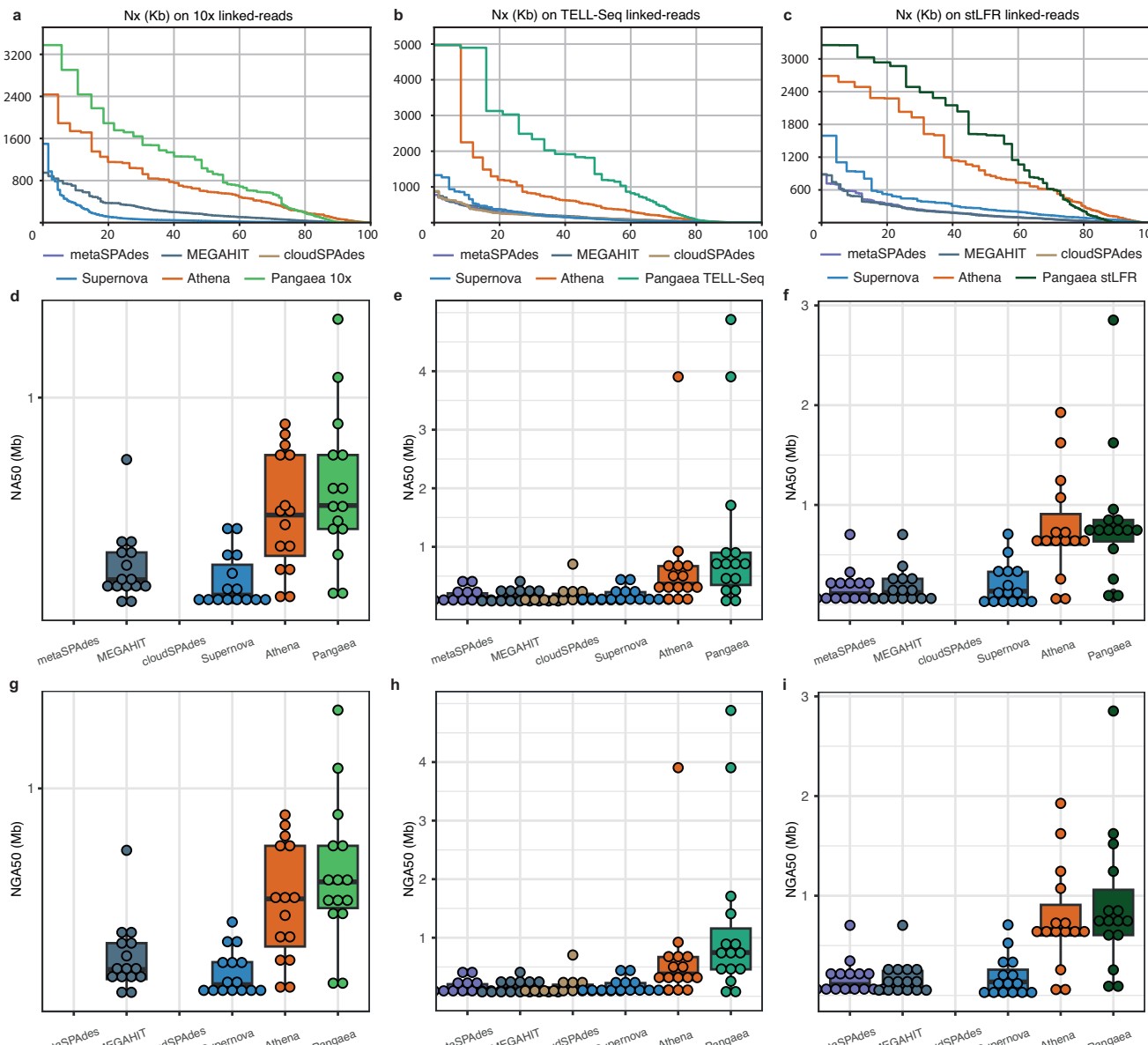

**Fig. 2 | Contig continuity of assemblies using barcode-removed short-reads (MEGAHIT, metaSPAdes) or linked-reads (cloudSPAdes, Supernova, Athena, Pangaea) from different linked-read sequencing technologies on ATCC-MSA-1003.** Nx, with x ranging from 0 to 100, assembled from 10x Genomics linked-reads (**a**), TELL-Seq linked-reads (**b**), and stLFR linked-reads (**c**) from ATCC-MSA-1003. NA50 for the 15 strains with abundances higher than 0.1% assembled from 10x Genomics linked-reads (**d**), TELL-Seq linked-reads (**e**), and stLFR linked-reads (**f**) on ATCC-MSA-1003. NGA50 for the 15 strains with abundances higher than 0.1% assembled from 10x Genomics linked-reads (**g**), TELL-Seq linked-reads (**h**), and stLFR linked-reads (**i**) on ATCC-MSA-1003. cloudSPAdes was unavailable for 10x Genomics linked-reads and stLFR linked-reads, as it requires extremely large memory (>2TB) on these datasets. The samples are biological replicates for (**d**–**i**), $n = 15$, each stands for a strain with abundance >0.1%. Box plots show the median (center line), 25th percentile (lower bound of box), 75th percentile (upper bound of box), and the minimum and maximum values within 1.5 × IQR (whiskers) as well as outliers (individual points). Source data are provided as a Source Data file.

## Pangaea outperforms the other assembly tools on the microbes with extremely low abundance

We employed ZYMO (Supplementary Table 4) to investigate the minimum abundance threshold necessary for identifying low-abundance microbes using Pangaea. Pangaea and Athena always generated better assemblies than the other tools on strains with different abundances (Supplementary Data 8). For the two strains (*Escherichia coli* and *Salmonella enterica*) with abundances between 0.01% and 0.1%, Pangaea achieved consistently higher NA50 (1.68 times on average) and NGA50 (1.68 times on average) than Athena (Supplementary Data 8). Although all assemblers could not generate contigs covering half of the genome of *Lactobacillus fermentum* (abundance 0.0089%), Pangaea still obtained the highest genome fraction (43.64%), which

was substantially higher than the values of the other linked-read assemblers (Supplementary Data 8). The strains with abundance below 0.001% were unable to be assembled by all assemblers (Supplementary Data 8).

## Pangaea generated high-quality assemblies on the human gut microbiomes

The assemblies generated by Pangaea achieved the highest total assembly lengths for all three samples (Table 2). Moreover, Pangaea achieved substantially higher N50s than all the other assemblers for both S1 (1.44 times of Athena, 1.06 times of Supernova, 3.65 times of cloudSPAdes, 4.71 times of MEGAHIT, 4.50 times of metaSPAdes; Table 2) and S2 (1.57 times of Athena, 2.58 times of Supernova, 6.33

**Table 2 | Assembly statistics for different assemblers using the barcode-removed linked-reads or linked-reads on human gut microbiomes**

| | Pangaea | Athena | Supernova | cloudSPAdes | MEGAHIT | metaSPAdes |
|---|---|---|---|---|---|---|
| **Human gut microbiome (S1)** | | | | | | |
| Total assembly length | **488,785,611** | 469,284,964 | 311,971,769 | 460,229,381 | 459,128,709 | 452,598,342 |
| Longest contig | 2,394,379 | 2,394,379 | **2,400,768** | 629,338 | 721,029 | 697,064 |
| Overall N50 | **64,394** | 44,759 | 60,619 | 17,651 | 13,670 | 14,325 |
| Overall N70 | 10537 | 10285 | **19660** | 5373 | 4856 | 5040 |
| Overall N90 | 2061 | 2050 | **2681** | 1717 | 1691 | 1725 |
| **Human gut microbiome (S2)** | | | | | | |
| Total assembly length | **408,819,148** | 393,685,495 | 290,599,879 | 378,182,799 | 381,215,035 | 374,166,135 |
| Longest contig | **2,877,256** | 1,903,088 | 1,152,844 | 538,650 | 480,921 | 443,089 |
| Overall N50 | **188,161** | 119,620 | 72,947 | 29,729 | 23,524 | 23,546 |
| Overall N70 | **40,808** | 36871 | 30,008 | 9,991 | 8,310 | 8,703 |
| Overall N90 | 2,858 | 2,933 | **3,976** | 2,078 | 2,034 | 2,099 |
| **Human gut microbiome (S3, contig > 1,000bps)** | | | | | | |
| Total assembly length | **323,679,273** | 270,721,423 | 222,438,535 | 284,200,415 | 301,723,162 | 298,374,739 |
| Longest contig | **4,769,781** | 4,570,813 | 1,384,668 | 526,768 | 526,790 | 592,456 |
| Overall N50 | 105,969 | **114,886** | 79,377 | 19,102 | 15,376 | 15,081 |
| Overall N70 | 10,283 | **28,002** | 25,965 | 4,674 | 4,841 | 4,545 |
| Overall N90 | 1,938 | **3,894** | 3,352 | 1,571 | 1,629 | 1,619 |
| **Human gut microbiome (S3, contig > 5,000bps)** | | | | | | |
| Total assembly length | **249,885,090** | 237,573,809 | 192,606,058 | 196,090,881 | 209,538,475 | 203,743,361 |
| Longest contig | **4,769,781** | 4,570,813 | 1,384,668 | 526,768 | 526,790 | 592,456 |
| Overall N50 | **275,648** | 157,810 | 105,541 | 44,589 | 33,264 | 36,343 |
| Overall N70 | **66,897** | 52,573 | 45,511 | 21,400 | 16,482 | 17,038 |
| Overall N90 | 10,958 | 12,471 | **13,132** | 8,165 | 7,527 | 7,545 |

The highest values are in bold.

times of cloudSPAdes, 8.00 times of MEGAHIT, 7.99 times of metaS-PAdes; Table 2). For S3, Pangaea generated much more sequences than the other assemblers (contig length > 1 kb; Methods), making it unfair to compare their N50 values directly. We transformed their assembly length to be comparable by removing contigs shorter than 5 kb. This led to a significant improvement in the contig N50 of Pangaea, which became the best one (Pangaea: 275.65 kb, Athena: 157.81 kb, Supernova: 105.54 kb, cloudSPAdes: 44.59 kb, MEGAHIT: 33.26 kb, metaS-PAdes: 36.34 kb; Table 2).

We grouped the contigs into MAGs and used NCMAGs (Methods) to evaluate the performance of metagenome assembly. Pangaea generated NCMAGs (Fig. 3a, e, i) of 24, 17, and 9 for S1, S2, and S3, which were much more than those generated by Athena, Supernova, cloudSPAdes, MEGAHIT and metaSPAdes. By calculating the number of NCMAGs at different minimum N50 thresholds, we found Pangaea obtained more NCMAGs than the other assemblers at almost all thresholds (Fig. 3b, f, j). Pangaea also outperformed the other assemblers with respect to the number of NCMAGs at different maximum read depth thresholds (Fig. 3c, g, k). Especially for the NCMAGs with N50s > 1 Mb, Pangaea achieved substantially more NCMAGs (S1: 8, S2: 4, S3: 5; Fig. 3d, h, l) than the other assemblers at all read depth thresholds, while Athena (the second best assembler) only produced 3, 1 and 2 NCMAGs on S1, S2 and S3, respectively (Fig. 3d, h, l).

**Pangaea generated high-quality assemblies for annotated microbes**

We annotated microbes of MAGs using Kraken2[34] with the NT database of NCBI (Methods). Total 61 microbes (S1: 26, S2: 19, S3: 16; shown in Fig. 4) were annotated from Pangaea's MAGs; 56 of them (S1: 24, S2: 16, S3: 16; Fig. 4) achieved the highest N50 and 33 microbes (S1: 16, S2: 8, S3: 9) had two-fold higher N50s comparing to the second best

assemblers (Fig. 4). Out of the 5 microbes for which Pangaea did not record the highest N50 values (as shown in Fig. 4), it produced N50s that were comparable to the best-performing assembler for *Alistipes indistinctus* and *Ruminococcus bicirculans* in S1, and for *Bacilli bacterium* in S2. Although Pangaea's N50 for *Roseburia hominis* was slightly lower, it achieved significantly greater genome completeness compared to Supernova (Pangaea: 96.54% completeness and 0.48% contamination; Supernova: 64.91% completeness and 0.00% contamination; Supplementary Data 9). Similarly, for *uncultured Clostridia* in S2, Pangaea's completeness was higher at 98.25% (contamination: 7.02%), surpassing Athena's completeness of 76.78% (contamination: 1.34%) (Supplementary Data 9).

Moreover, Pangaea generated more NCMAGs for the annotated microbes. There were 13 microbes (S1: 7, S2: 2, S3: 4; Fig. 4) that could be uniquely assembled as NCMAGs by Pangaea, whereas all the other assemblers either produced at lower quality or failed to assemble corresponding MAGs. Pangaea outperformed other assemblers in analyzing three human gut microbiomes by identifying 17 annotated microbes from NCMAGs with N50 values exceeding 1 Mb. In contrast, Athena identified 6, Supernova found 1, and cloudSPAdes, MEGAHIT, and metaSPAdes did not produce any microbes meeting this benchmark (Fig. 4). In addition, Pangaea recognized 6 unique microbes (*Bacteriophage* sp. and *Dialister* from S1, and *uncultured bacterium*, *Muribaculum gordoncarteri*, *Parabacteroides* and *Prevotella copri* from S2) that were not found by any other assemblers, and *Dialister* from S1 was represented by a NCMAG (Fig. 4).

**Strong collinearities between NCMAGs and their closest reference genomes**

We compared the NCMAGs that can be annotated as species with their closest reference genomes to evaluate their collinearities (Methods).

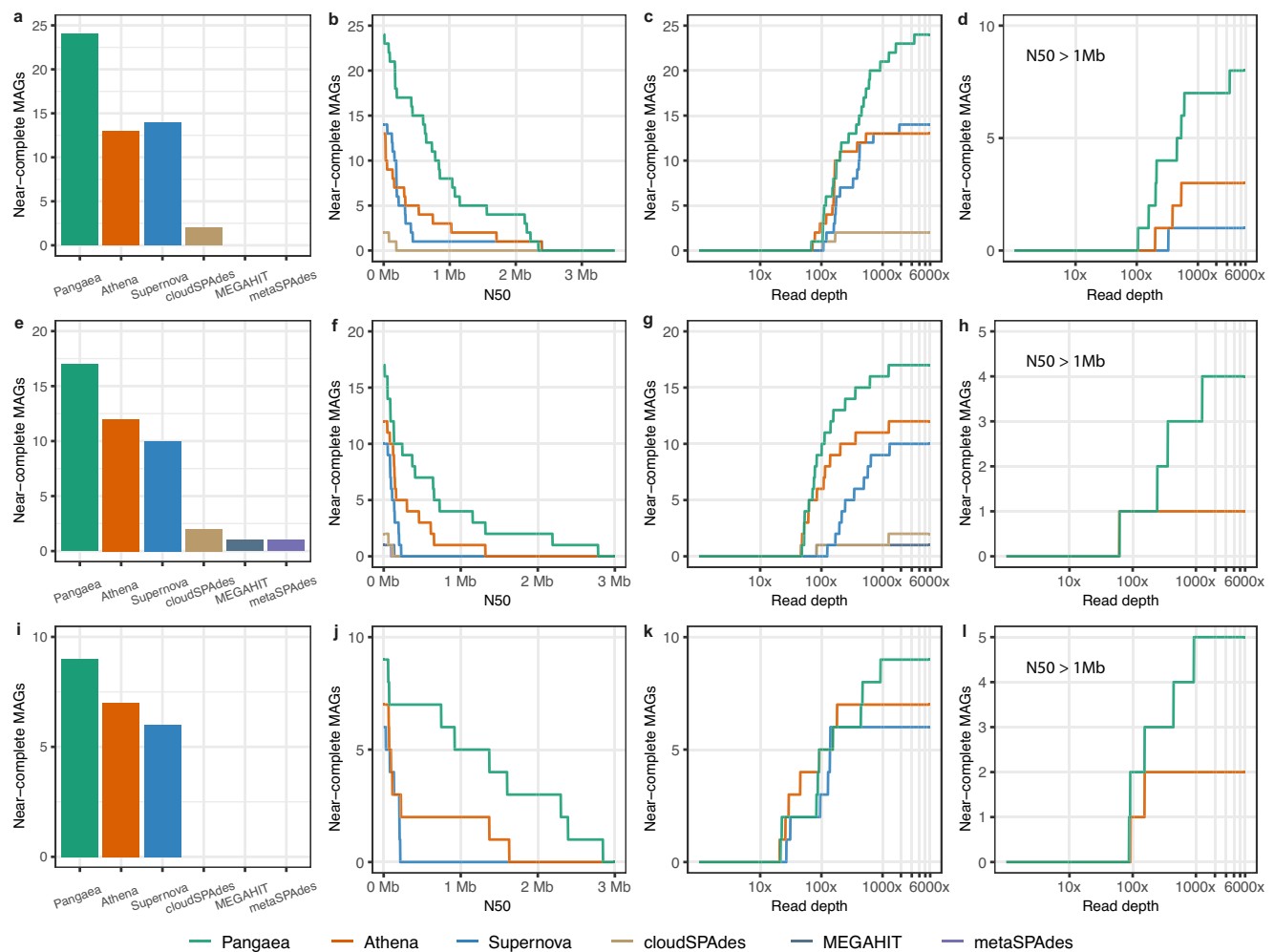

**Fig. 3 | The number of NCMAGs from the three human gut microbiomes under different N50s and read depths.** The number of NCMAGs for S1 (**a**), the number of NCMAGs by thresholding minimum N50 for S1 (**b**), the number of NCMAGs by thresholding maximum read depth for S1 (**c**), and the number of NCMAGs with N50 > 1Mb by thresholding maximum read depth for S1 (**d**). The number of NCMAGs for S2 (**e**), the number of NCMAGs by thresholding minimum N50 for S2 (**f**), the number of NCMAGs by thresholding maximum read depth for S2 (**g**), and the number of NCMAGs with N50 > 1Mb by thresholding maximum read depth for S2 (**h**). The number of NCMAGs for S3 (**i**), the number of NCMAGs by thresholding minimum N50 for S3 (**j**), the number of NCMAGs by thresholding maximum read depth for S3 (**k**), and the number of NCMAGs with N50 > 1Mb by thresholding maximum read depth for S3 (**l**). Source data are provided as a Source Data file.

The NCMAGs generated by different assemblers and their closest reference genomes had comparable average alignment identities (Pangaea: 98.16%, Athena: 98.12%, Supernova: 98.31%, cloudSPAdes: 98.77%, MEGAHIT: 98.7%, metaSPAdes: 98.8%) and average alignment fractions (Pangaea: 87.9%, Athena: 88.6%, Supernova: 88.4%, cloudSPAdes: 88.5%, MEGAHIT: 88%, metaSPAdes: 90%; Supplementary Data 10), while Pangaea produced significantly more species-level NCMAGs than the other assemblers (Pangaea: 29, Athena: 21, Supernova: 14, cloudSPAdes: 2, MEGAHIT: 1, metaSPAdes: 1; Supplementary Data 10).

The NCMAGs assembled by Pangaea with species-level annotations had high collinearities with their closest reference genomes (Fig. 5; Supplementary Fig. 4). Some of these NCMAGs showed inversions and rearrangements in comparison to the reference sequences, including *Alistipes communis* (S1; Supplementary Fig. 4), *Desulfovibrio desulfuricans* (S1; Supplementary Fig. 4) and *Siphoviridae* sp. (S2; Fig. 5g). Pangaea assembled NCMAGs for *Siphoviridae* sp. from both S2 and S3 (Fig. 5g, h). The two NCMAGs had comparable total sequence lengths (S2: 2.20 Mb and S3: 2.39 Mb; Supplementary Data 9) and better N50 was achieved in S3 (N50: 1.16 Mb for S2 and 2.39 Mb for S3; Supplementary Data 9). Note that the NCMAG of *Siphoviridae* sp. from

S3 is a single-contig NCMAG, but it's not a circular contig which might be because it does not include all single-copy genes (completeness: 91.28%; Supplementary Data 9).

Pangaea could generate NCMAGs that count be annotated as species with better quality and larger N50 values than the MAGs for the same species generated from other assemblers, such as *Alistipes* sp. from S1 and *Collinsella aerofaciens* from S2 (Fig. 5d, e; Supplementary Data 9). The evaluation of the read depths and GC-skew of the MAGs revealed that Pangaea recovered the regions with extremely low read depths and high GC-skew, such as the region at ~480 kb of *Faecalibacterium prausnitzii* from S3 (Supplementary Fig. 4). This indicates Pangaea has the potential to reconstruct hard-to-assemble genomic regions.

## Pangaea generated complete and circular MAGs using linked-reads

We examined if there existed complete and circular genomes in NCMAGs based on the circularization module in Lathe[11] (Methods). We found that only Pangaea generated three circular NCMAGs, which were annotated as *Bifidobacterium adolescentis* (S1), *Myoviridae* sp. (S1) and *Wujia chipingensis* (S3). For each of the three microbes, Pangaea

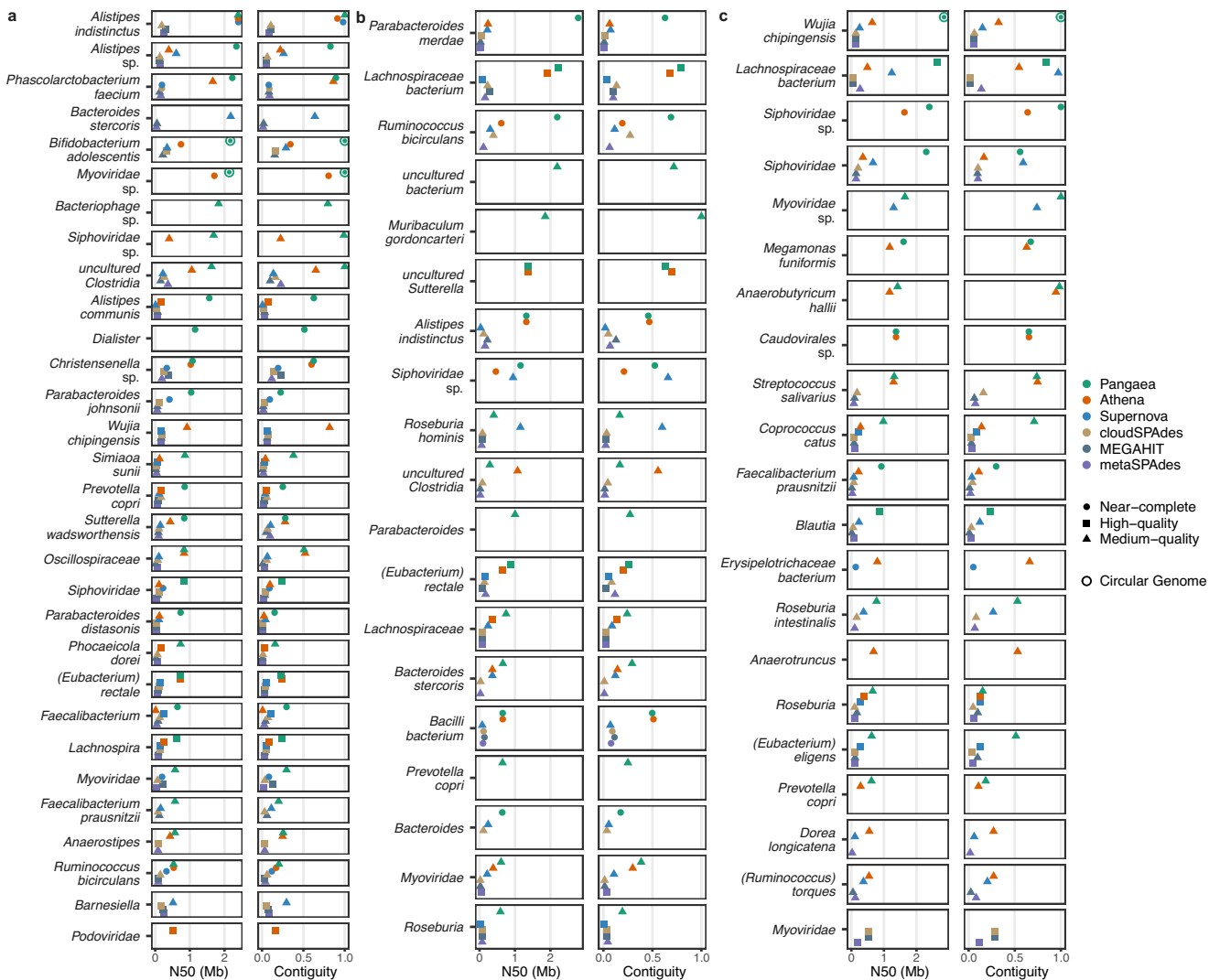

**Fig. 4 | The annotated microbes for the MAGs generated by different assemblers from the three human gut microbiomes. a** The annotated microbes produced by Pangaea, Athena, Supernova, cloudSPAdes, MEGAHIT, and metaSPAdes from S1 with N50 and contiguity (defined as N50/draft genome length). **b** The annotated microbes produced by Pangaea, Athena, Supernova, cloudSPAdes, MEGAHIT, and metaSPAdes from S2 with N50 and contiguity. **c** The annotated microbes produced by Pangaea, Athena, Supernova, cloudSPAdes, MEGAHIT, and metaSPAdes from S3 with N50 and contiguity. The microbes were shown if the N50s of their MAGs were larger than 500 kb by any assembler. If the same microbe was annotated by more than one MAG from the same assembler, the one with the highest N50 was selected. Source data are provided as a Source Data file.

produced a gapless contig with perfect collinearity with the closest reference genomes (Fig. 5a, b, i).

Athena generated three and two contigs for *B. adolescentis* (from S1) and *Myoviridae* sp. (from S1), with substantially lower contig N50 values than the contigs obtained by Pangaea for these two species (*B. adolescentis*: Pangaea = 2.17 Mb, Athena = 744.54 kb; *Myoviridae* sp.: Pangaea = 2.14 Mb, Athena = 1709.63 kb; Supplementary Data 9). cloudSPAdes obtained a high-quality MAG for *B. adolescentis*, but this MAG could not be annotated with any rRNAs, and the contig continuity was much lower than the corresponding MAG from Pangaea (Pangaea = 2.17 Mb, cloudSPAdes = 329.67 kb). Supernova, MEGAHIT, and metaSPAdes could only generate incomplete MAGs or could not assemble these two species, and the completeness of their candidate MAGs was significantly lower than that of MAGs generated by Pangaea (Supplementary Data 9). For *W. chipingensis* from S3, Pangaea was the only assembler that obtained NCMAG and got a significantly higher contig N50 than the other assemblers (Pangaea: 2.85Mb, Athena: 639.65 kb, Supernova: 254.31 kb, cloudSPAdes: 135.56 kb, MEGAHIT: 134.14 kb, metaSPAdes: 146.44 kb; Supplementary Data 9).

**Investigate the impact of sequencing depth on assembly results**

We assessed Pangaea's effectiveness on datasets with varying sequencing depths by generating subsets of linked-reads at 5 Gb, 10 Gb, 20 Gb, 50 Gb, and 100 Gb from ATCC-MSA-1003 (TELL-Seq), CAMI-high (stLFR), and S1 (stLFR) (Methods). We then compared Pangaea's assembly performance on these subsampled read sets against other assembly algorithms. Increasing the sequencing depth improved the assembly quality of metagenomes across all three microbial community datasets for all involved assemblers (Supplementary Data 11).

Pangaea demonstrated superior performance compared to the other assemblers across all subsampling datasets. Pangaea obtained better overall NA50 and NGA50 per strain than the second-best assembler on all subsampled datasets from ATCC-MSA-1003 (Supplementary Data 11). Pangaea achieved better NA50 values than the other tools except for Athena on CAMI-high. On datasets under 50 Gb from CAMI-high, Athena achieved better overall NA50s; however, this higher continuity came at the cost of significantly shorter total assembly lengths (Supplementary Data 11). On the dataset of 100Gb

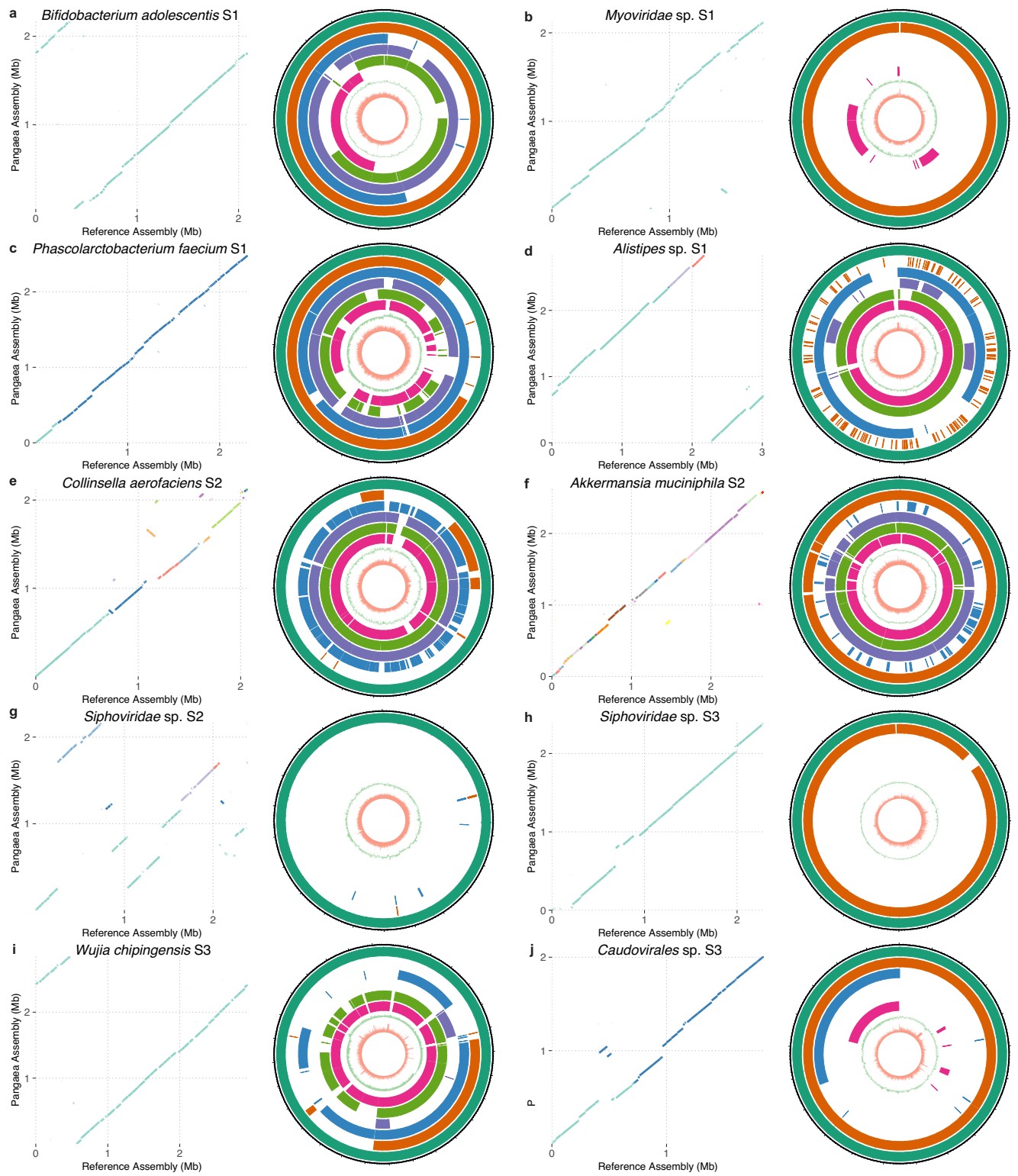

**Fig. 5 | Genome collinearity between the species-level NCMAGs produced by Pangaea and their closest reference genomes (dot plots), and comparison of different MAGs of the same species (circos plots). a–d** The selected NCMAGs of Pangaea for S1. **e–g** The selected NCMAGs of Pangaea for S2. **h–j** The selected NCMAGs of Pangaea for S3. Pangaea obtained complete and circular NCMAGs for (**a**, **b**, **i**). Concentric rings in the circos plots from outermost to innermost display Pangaea (dark green), Athena (orange), Supernova (blue), cloudSPAdes (purple), MEGAHIT (light green), metaSPAdes (pink), GC-skew, and read depth of Pangaea MAGs, respectively. Long tick marks on the outer black circle indicate 100 kb intervals. For species with multiple MAGs from the same assembler, only the MAG with the highest N50 is shown. The remaining species-level NCMAGs of Pangaea are shown in Supplementary Fig. 4. Source data are provided as a Source Data file.

from CAMI-high, Pangaea achieved the highest overall NA50, NGA50 per strain, and the number of non-zero NGA50s (Supplementary Data 11).

Pangaea consistently generated a higher N50 than Athena on all the subsampled datasets on S1. On the 5Gb and 10 Gb datasets of S1, Supernova achieved the highest N50 values, but this was accompanied by the shortest assembled sequence lengths relative to the other assemblers (Supplementary Data 11). Pangaea significantly outperformed Athena on the 100Gb dataset of S1 with respect to the number of NCMAGs (Pangaea = 18, Athena = 10), while they showed similar performance on the other subsampled datasets. These findings indicate that for human gut microbiomes, increased sequencing depth improves Pangaea's performance in producing NCMAGs. Nevertheless, it should be emphasized that there is a positive relationship between data volume and the performance of all assemblers in generating NCMAGs. This may be because of the high PCR duplication rate of linked-read sequencing technologies[17], where 61.73% of stLFR linked-reads in S1 were marked as duplication (Methods). Therefore, the sequencing amount of linked-reads used for metagenome assembly is commonly higher compared to short-reads to ensure sufficient informative reads are given.

## Pangaea generated high-quality assemblies on short-reads with virtual barcodes from long-reads

As long-read sequencing has been limited in the applications in large-scale cohorts due to its high cost[12], the hybrid assembly combining short-reads (high depth) and long-reads (typically with shallow depth) in assembly was proposed as a cost-effective way to produce high-quality assemblies[15]. We evaluated the generalizability of Pangaea on hybrid assembly by attaching virtual barcodes to short-reads using shallow depth long-reads (Methods). We removed the barcodes of TELL-Seq (ATCC-MSA-1003), stLFR (ZYMO), and stLFR (CAMI-high) linked-reads, and added virtual barcodes to the short-reads from alignments against long-reads of PacBio or Oxford Nanopore (Supplementary Table 2; Methods). We adjusted the workflow of Pangaea to integrate with the existing hybrid assemblers using both short- and long-reads, including MetaPlatanus (Pangaea-MetaPlatanus), hybridSPAdes (Pangaea-Hybridspades), and OPERA-MS (Pangaea-Operams) (Methods).

We compared Pangaea-MetaPlatanus, Pangaea-Hybridspades and Pangaea-Operams with their original hybrid assemblers as well as Athena (for linked-reads) and metaSPAdes (for short-reads), and observed that virtual barcodes could prominently increase contig N50, overall NA50 and average NGA50 per strain on ATCC-MSA-1003 and CAMI-high (Table 3; Fig. 6). All assemblers could not assemble with high genome fractions (>50%) for most of the 5 strains with the lowest abundance (0.02%) in ATCC-MSA-1003. However, Pangaea could generate more sequences for the informative long contigs (>10 kb) than the corresponding hybrid assembly tools (Supplementary Data 12).

We evaluated the capability of Pangaea to detect the extremely low-abundance microbes in ZYMO. For the two strains with abundances >1%, Pangaea-Hybridspades generated much higher NGA50 than Hybridspades (2.23 times); the performance of Pangaea-MetaPlatanus and Pangaea-Operams was comparable with their original tools; this is acceptable as ZYMO is a simple metagenome where the existing hybrid assemblers can already assembly well for those high-abundance microbes in this dataset (Supplementary Data 12). For the two strains with abundance between 0.1% and 1%, Pangaea-Hybridspades and hybridSPAdes had comparable NGA50s, while Pangaea-MetaPlatanus (1.18 times) and Pangaea-Operams (1.45 times) produced better average NGA50 than MetaPlatanus and OPERA-MS, respectively (Supplementary Data 12). For the other two strains with abundance <0.1%, Pangaea-Operams still obtained substantially higher average NGA50 than Operams (2.37 times), and all of the three models

from Pangaea generated more sequences for long contigs (>10 kb) than their original tools, respectively (Supplementary Data 12). All assemblers produced low-quality assemblies for the strain with an abundance <0.01% (Supplementary Data 12).

## Evaluation of running time and maximum memory usage

We compared the computational performance (CPU time, Real time, and Maximum Resident Set Size [RSS]) of the benchmarked assemblers on ZYMO (Fig. 7; Methods). Both Athena and Pangaea required the assemblies from the other tools, so we only considered their additional processing time and memory. MEGAHIT was the fastest assembler with the lowest maximum RSS (Fig. 7). metaSPAdes and cloudSPAdes consumed substantially higher CPU times than the other assemblers (Fig. 7a). The real time (wall clock time) used by metaSPAdes, cloudSPAdes, Supernova, Athena, hybridSPAdes, and MetaPlatanus were comparable and significantly higher than those consumed by MEGAHIT, Pangaea, OPERA-MS, Pangaea-Operams and Pangaea-Hybridspades (Fig. 7b). MetaPlatanus and Supernova used the highest maximum RSS on ZYMO (Fig. 7c). The maximum RSSs used by metaSPAdes, cloudSPAdes, and hybridSPAdes were close to each other, and much higher than those needed by the other assemblers except for MetaPlatanus and Supernova (Fig. 7c). These results revealed Pangaea could improve the assembly quality of the existing assemblers in a reasonable time and using a relatively low maximum RSS.

## Discussion

Short-read sequencing has proven to be an important approach for analyzing human gut microbiota from large sequencing cohorts. However, its lack of long-range DNA connectivity makes assembling conserved sequences, intra- and inter-species repeats, and ribosomal RNAs (rRNAs) difficult[5]. It has limitations in producing complete microbial genomes and long-read sequencing is relatively expensive to be applied to large cohorts. Cost-effective linked-read sequencing technologies, which attach barcodes to short-reads to provide long-range DNA connectivity, have achieved great success in improving contig continuity in metagenome assembly[14,19]. Unlike 10x Genomics linked-reads, stLFR[17] and TELL-Seq linked-reads[18] have high barcode specificity, but a dedicated assembler that could make full use of this characteristic to improve metagenome assembly is lacking. Besides linked-reads, the long-range connectivity of short-reads could also be provided by virtual barcodes from other independent sequencing technologies (e.g., long-reads).

In this study, we developed Pangaea to improve metagenome assembly by leveraging long-range connectivity from linked-reads and long-reads. It considers the co-barcoded reads as long DNA fragments and extracts their $k$-mer histograms and TNFs for co-barcoded read binning. This strategy significantly reduces the complexity of metagenomic sequencing data and makes the assembly more efficient. Because sequence clustering is sensitive to data sparsity and noise, Pangaea represents the input features in a low-dimensional latent space using VAE. We also designed a weighted sampling strategy to generate a balanced training set for microbes with different abundances. This module primarily advantages microbes with high- and medium-abundance, because they are more robust to mis-binning and usually have sufficient data for assembly in the corresponding bins. Pangaea adopts a multi-thresholding reassembly strategy to rescue the reads from low-abundance microbes. It eliminates short-reads from high-abundance microbes in the assembly graph gradually to differentiate the assembly graph structures from low-abundance microbes and sequencing errors. In the third module, we merged the assemblies from different strategies due to their complementary nature of each other.

A previous study[35] showed that co-assembly with multiple samples could improve completeness and decrease the contamination of MAGs. However, the co-assembly of metagenomes derived from large

**Table 3 | Assembly statistics for different assemblers using short-reads, short-reads with virtual barcodes, or short- and long-reads on mock communities**

| | Total assembly length | Genome fraction (%) | Largest alignment | Overall N50 | Overall N70 | Overall N90 | Overall NA50 | NA50 per strain | NGA50 per strain | Non-zero NGA50 |
|---|---|---|---|---|---|---|---|---|---|---|
| **ATCC-MSA-1003** | | | | | | | | | | |
| Pangaea-MP | **61,826,844** | 77.21 | **4,613,760** | 419,529 | 123,550 | 6,828 | 227,880 | **551,174** | **599,918** | **16** |
| MetaPlatanus | 50,420,118 | 65.78 | 4,143,574 | 232,150 | 64,539 | 2,484 | 207,981 | 342,514 | 338,045 | 12 |
| Pangaea-HS | 61,432,598 | 82.69 | 2,824,147 | **475,582** | **150,196** | 6,824 | **338,892** | 410,176 | 412,875 | **16** |
| hybridSPAdes | 60,794,505 | **82.93** | 1,518,615 | 255,665 | 88,668 | 6,333 | 236,258 | 229,866 | 229,492 | **16** |
| Pangaea-OP | 53,145,042 | 63.25 | 3,027,146 | 345,524 | 108,841 | 4,801 | 236,263 | 219,172 | 311,080 | **16** |
| OPERA-MS | 44,017,454 | 57.87 | 885,414 | 55,544 | 19,419 | 2,919 | 49,863 | 104,942 | 105,557 | 11 |
| Athena | 60,077,880 | 81.57 | 1,175,563 | 176,953 | 81,610 | **7,766** | 173,018 | 158,550 | 157,166 | **16** |
| metaSPAdes | 60,648,311 | 82.46 | 776,102 | 112,342 | 49,466 | 5,429 | 105,630 | 119,253 | 118,391 | **16** |
| **CAMI-high** | | | | | | | | | | |
| Pangaea-MP | 790,047,911 | 27.83 | 3,200,312 | **193,286** | **92,221** | **22,625** | **179,405** | **66,510** | **47,460** | 187 |
| MetaPlatanus | 759,582,683 | **28.28** | 2,525,754 | 146,524 | 66,065 | 11,900 | 143,981 | 55,924 | 39,696 | 187 |
| Pangaea-HS | 765,341,275 | 26.99 | 2,552,627 | 185,706 | 86,908 | 22,576 | 172,580 | 57,562 | 43,844 | 180 |
| hybridSPAdes | 759,703,702 | 27.04 | 3,195,687 | 148,522 | 68,335 | 14,950 | 140,963 | 50,524 | 39,710 | 179 |
| Pangaea-OP | 790,451,622 | 27.40 | **3,370,333** | 172,800 | 78,129 | 16,150 | 160,739 | 60,235 | 45,323 | 181 |
| OPERA-MS | **797,297,690** | 27.01 | 1,860,977 | 95,314 | 41,765 | 8,017 | 94,811 | 34,746 | 21,890 | 174 |
| Athena | 738,341,840 | 26.27 | 2,520,898 | 151,113 | 72,087 | 17,299 | 149,290 | 49,612 | 36,428 | 173 |
| metaSPAdes | 759,108,499 | 27.01 | 2,520,915 | 133,877 | 62,467 | 12,521 | 132,603 | 45,261 | 34,551 | 177 |
| **ZYMO** | | | | | | | | | | |
| Pangaea-MP | **36,160,031** | 48.66 | 2,719,840 | **444,651** | 181,141 | 73,012 | **345,995** | 391,840 | 392,681 | 6 |
| MetaPlatanus | 35,906,273 | **48.91** | 2,423,776 | 250,282 | 137,973 | 50,299 | 242,683 | 429,587 | 428,086 | 6 |
| Pangaea-HS | 35,806,901 | 48.47 | 2,350,555 | 444,276 | **203,542** | **76,164** | 342,859 | 697,544 | 704,255 | 6 |
| Hybrid-SPAdes | 35,517,607 | 48.63 | 1,799,936 | 289,924 | 154,423 | 69,345 | 287,773 | 387,101 | 383,552 | 6 |
| Pangaea-OP | 35,917,521 | 48.62 | **3,579,300** | 347,153 | 156,102 | 59,828 | 301,599 | **756,021** | **757,488** | 6 |
| OPERA-MS | 35,334,818 | 48.28 | 3,579,284 | 131,207 | 70,061 | 21,837 | 124,358 | 683,735 | 681,358 | 6 |
| Athena | 34,567,225 | 47.34 | 1,282,835 | 237,784 | 126,002 | 59,781 | 232,219 | 214,809 | 210,268 | 6 |
| metaSPAdes | 35,511,020 | 48.46 | 847,644 | 191,688 | 106,796 | 44,425 | 191,688 | 183,935 | 179,371 | 6 |

Pangaea-MP stands for Pangaea-MetaPlatanus, Pangaea-HS stands for Pangaea-hybridSPAdes and Pangaea-OP stands for Pangaea-Operams.
The highest values are in bold.

samples poses a practical challenge due to time and computational limitations. Because the sequencing data from all the individual samples need to be merged before co-assembly. Read binning is a sophisticated strategy that enables co-assembly on large datasets by producing smaller and simpler subsets of reads, thereby facilitating the assembly process. Several studies have attempted to apply read binning to short-read metagenomic sequencing[26–28], but it is exceedingly difficult in practice. The fragments of short-reads are too short to allow the extraction of stable sequence abundance and composition features from the individual reads. Existing read binning tools have to identify the overlap between each pair of reads for binning. However, millions or even billions of short-reads make the overlap-based read binning algorithm extremely slow and highly memory intensive. Overlap Graph-based Read clustEring (OGRE)[26] was developed to improve the computational performance of read binning, but it still consumed 2263 CPU hours even for the low-complexity dataset of CAMI[26]. We evaluated OGRE on stLFR linked-reads of ATCC-MSA-1003 (664.77 M read pairs) and observed that OGRE crashed due to insufficient memory if 100 threads were applied. If fewer threads were applied, the binning time would become extremely long (more than 2 weeks). Pangaea with 100 threads only took 64.06 h in real time, 514.63 h in CPU time, and consumed 281.99 GB of maximum RSS to group and assemble this linked-reads dataset.

We evaluated the number of misassemblies, the number of intra-species translocations (chimeric assemblies), and sequence identities ((Total_Aligned_Length-Mismatches)/Total_Aligned_Length) from the MetaQUAST reports of the three mock communities (ATCC-MSA-1003, ZYMO, and CAMI-high; Supplementary Data 13). As an ensemble assembler, the assemblies of Pangaea were observed to have more misassemblies than those of Athena. This is because Pangaea integrated contigs from both co-barcoded read binning and multi-thresholding reassembly to improve the assembly continuity and induced some misassemblies during the merging process. However, Pangaea still achieved much higher NA50 and NGA50 per strain by breaking contigs at misassemblies (Tables 1, 2, 3) on most of the datasets. Moreover, despite the misassemblies, Pangaea was still able to achieve the highest number of NCMAGs and even complete and circular genomes in real complex microbiomes. In this study, we used three metagenomic sequencing datasets from female fecal samples to evaluate our method. However, given the thorough testing on mock communities, including ATCC-MSA-1003, ZYMO, and CAMI-high, we are confident that our approach can be applied to metagenomic data from different sources.

Long-read sequencing has received increasing attention due to its ability to generate complete microbial genomes from complex communities. However, it is limited by a relatively high cost for large-cohort studies. In contrast, linked-read and hybrid sequencing (deep short-read and shallow long-read sequencing) techniques are cost-effective. Linked-read sequencing only requires a tiny amount of input DNA, and can thus be a complementary solution to long-read sequencing. In our experiments, we found that long-read assemblies

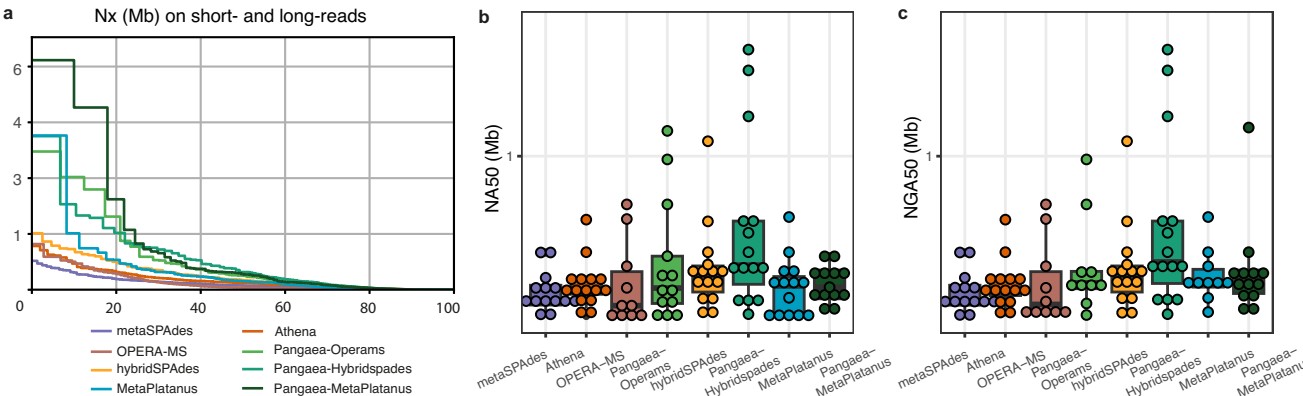

**Fig. 6 | Contig continuity of different assemblies using short-reads, short-reads with virtual barcodes, or short- and long-reads on ATCC-MSA-1003. a** Nx, with x ranging from 0 to 100 of different assemblies using short-reads (metaSPAdes), short-reads with virtual barcodes (Athena, Pangaea-Operams, Pangaea-Hybrid-spades, and Pangaea-MetaPlatanus), and using short- and long-reads (OPERA-MS, hybridSPAdes, and MetaPlatanus) on ATCC-MSA-1003. **b** NA50 of the 15 strains with abundance >0.1% assembled from short-reads (metaSPAdes), short-reads with virtual barcodes (Athena, Pangaea-Operams, Pangaea-Hybridspades, and Pangaea-MetaPlatanus), and from short- and long-reads (OPERA-MS, hybridSPAdes, and MetaPlatanus) on ATCC-MSA-1003. **c** NGA50 of the 15 strains with abundance >0.1% assembled from short-reads (metaSPAdes), short-reads with virtual barcodes (Athena, Pangaea-Operams, Pangaea-Hybridspades, and Pangaea-MetaPlatanus), and from short- and long-reads (OPERA-MS, hybridSPAdes, and MetaPlatanus) on ATCC-MSA-1003. The samples are biological replicates for (**b–c**), $n = 15$, each stands for a strain with abundance >0.1%. Box plots show the median (center line), 25th percentile (lower bound of box), 75th percentile (upper bound of box), and the minimum and maximum values within $1.5 \times$ IQR (whiskers) as well as outliers (individual points). Source data are provided as a Source Data file.

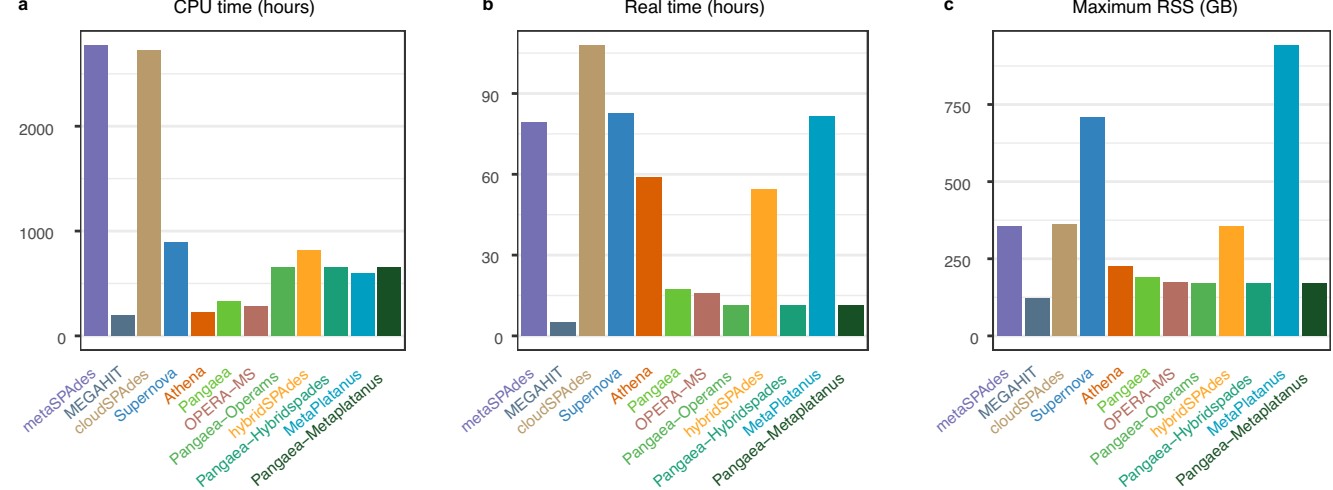

**Fig. 7 | The CPU time, real time and maximum resident set size (RSS) of different assemblers on ZYMO mock community. a** CPU times for assemblers using barcode-removed short-reads (metaSPAdes, MEGAHIT), linked-reads (cloud-SPAdes, Supernova, Athena, Pangaea), short-reads with virtual barcodes (Pangaea-Operams, Pangaea-Hybridspades, and Pangaea-MetaPlatanus), and short- and long-reads (OPERA-MS, hybridSPAdes, and MetaPlatanus). **b** Real times for assemblers using barcode-removed short-reads (metaSPAdes, MEGAHIT), linked-reads (cloudSPAdes, Supernova, Athena, Pangaea), short-reads with virtual barcodes (Pangaea-Operams, Pangaea-Hybridspades, and Pangaea-MetaPlatanus), and short- and long-reads (OPERA-MS, hybridSPAdes, and MetaPlatanus). **c** Maximum RSS for assemblers using barcode-removed short-reads (metaSPAdes, MEGAHIT), linked-reads (cloudSPAdes, Supernova, Athena, Pangaea), short-reads with virtual barcodes (Pangaea-Operams, Pangaea-Hybridspades, and Pangaea-MetaPlatanus), and short- and long-reads (OPERA-MS, hybridSPAdes, and MetaPlatanus). Source data are provided as a Source Data file.

had 60.98% fewer NCMAGs than linked-read assemblies from Pangaea (Supplementary Note 4) after contig binning by MetaBat2, indicating that some microbes might be lost due to insufficient long-read sequencing depth. Even the state-of-the-art tool VAMB[36] was used for contig binning, the number of NCMAGs from long-read (NCMAG: S1 = 6, S2 = 6; assembled by metaFlye) and short-read assemblies (NCMAG: S1 = 0, S2 = 0, assembled by metaSPAdes; S1 = 1, S2 = 0, assembled by MEGAHIT) remained substantially lower than linked-read assembly (NCMAG: S1 = 21, S2 = 21; assembled by Pangaea). Similar observations have been reported in a previous study[11]. Although stLFR and TELL-Seq linked-reads had high barcode specificity in ATCC-MSA-1003, we observed that a considerable fraction of

barcodes still contained more than one fragment (stLFR = 37.02%, TELL-Seq = 72.95%), which could complicate the deconvolution of barcodes for the existing linked-read assemblers. We believe that further protocol improvement for these technologies (e.g., increasing the number of beads) may further improve their metagenome assembly performance.

## Methods

The study complies with all relevant ethical regulations and was approved by the Ethics Committee of BGI (BGI-IRB 20145). We have received written informed consent from the human participants in this study.

### DNA preparation and sequencing for linked-read sequencing

On ATCC-MSA-1003, the microbial DNAs were extracted directly from the 20 Strain Staggered Mix Genomic Material (ATCC MSA-1003) without size selection using a QIAamp DNA stool mini kit (Qiagen, Valencia, CA, USA). For the human gut microbiomes, microbial DNAs from stool samples of three individuals (S1, S2, and S3) were extracted using the QIAamp DNA stool mini kit (Qiagen) and size-selected using a BluePippin instrument targeting the size range of 10−50 Kb according to the manufacturer's protocol. The three individuals were female, with ages 36, 32, and 30, respectively. Sex or gender of participants was determined based on self-report. No sex- and gender-based analyses was performed, this was because the relation between human gut microbiomes and the gender was beyond the scope of this study. Participant compensation was not applied. The stLFR libraries were prepared using the stLFR library prep kit (16 RXN), followed by 2 × 100 paired-end short-read sequencing using BGISEQ-500. The TELL-Seq library for ATCC-MSA-1003 was prepared using the TELL-Seq™ Library Prep Kit, followed by 2 × 146 paired-end sequencing on an Illumina sequencing system. The Shannon diversities of all datasets were calculated using MetaPhlAn (v4.0.6)[37] with default parameters. PCR duplication rates of linked-reads were reported by fastp (v0.21.0)[38] with default parameters. Subsampling for linked-reads was performed by sampling the depth of long fragments, i.e., reads with the same barcode are either all kept or all discarded in the subsampled dataset.

### Simulate stLFR linked-reads for ZYMO and CAMI-high

We downloaded the reference genomes of the 10 strains included in ZymoBIOMICS™ Microbial Community Standard II (Log Distribution)[33] (ZYMO) and used LRTK (v1.7)[39] to simulate stLFR linked-reads with the same strain composition as the ZYMO mock community. For CAMI-high, we used the microbial composition from the first sample of the five time-series samples provided by CAMI Challenge Dataset CAMI_high[32] and simulated stLFR linked-reads using LRTK (v1.7)[39]. The Nanopore long-reads of CAMI-high was simulated using CAMISIM (v1.2-beta)[40].

### Generate virtual barcodes of short-reads from long-reads

We attached the same long-read indexes as virtual barcodes to short-reads if they were aligned to the same long-read. We mapped the short-reads to long-reads using BWA-MEM (v0.7.17)[41] and removed the spurious alignments if the minimum aligned nucleotides were below 60bps. If a short-read was aligned to multiple long-reads, we randomly chose one of the long-read indexes as its barcode.

### Extract *k*-mer histogram and TNFs from co-barcoded reads

We extracted *k*-mer histograms and TNFs from the co-barcoded reads if their total lengths were longer than 2 kb to ensure feature stability. A *k*-mer histogram was calculated based on global *k*-mer occurrences and could reflect the abundance features of the microbial genome[42]. We adopted $k = 15$ as used in the previous studies[42,43], and calculated the global 15-mer frequencies using the whole dataset. We removed 15-mers with frequencies higher than 4000 (to avoid repetitive sequences) and divided the global frequency distribution into 400 bins with equal sizes (the $i^{th}$ bin denoted frequencies between $10*i − 10$ and $10*i$). We collected the co-barcoded reads for each barcode and divided these reads into 15-mers, which were assigned to the 400 bins based on their global frequencies. We calculated the number of 15-mers allocated to each bin and generated a count vector with 400 dimensions as the *k*-mer histogram of the specific barcode. A TNF vector was constructed by calculating the frequencies of all 136 non-redundant 4-mers from co-barcoded reads. The *k*-mer histogram and TNF vector were normalized to eliminate the bias introduced by the different lengths of co-barcoded reads.

### Binning co-barcoded reads with a VAE

The normalized *k*-mer histogram ($X_A$) and TNF vector ($X_T$) were concatenated into a vector with 536 dimensions as the input to a VAE

(Fig. 1b; Supplementary Note 2; the use of VAE was inspired by ref. 36). The encoder of VAE consisted of two fully connected layers with 512 hidden neurons, and each layer was followed by batch normalization[44] and a dropout layer ($P = 0.2$)[45]. The output of the last layer was fed to two parallel latent layers with 32 hidden neurons for each to generate $\mu$ and $\sigma$ for a Gaussian distribution $\mathcal{N}(\mu, \sigma^2)$, from which the embedding $Z$ was sampled. The decoder also contained two fully connected hidden layers of the same size as the encoder layers to reconstruct the input vectors ($\hat{X}_A$ and $\hat{X}_T$) from the latent embedding $Z$. We applied the *softmax* activation function on $\hat{X}_A$ and $\hat{X}_T$ to achieve the normalized output vectors, because the input features $X_A$ and $X_T$ were both normalized. The loss function (*Loss*) was defined as the weighted sum of three components: the reconstruction loss of *k*-mer histogram ($L_A$), the reconstruction loss of TNF vectors ($L_T$), and the Kullback-Leibler divergence loss ($L_{KL}$) between the latent and the prior standard Gaussian distributions. We adopted cross-entropy loss for $L_A$ and $L_T$ to deal with probability distributions and formularized the loss terms as follows:

$$L_A = \sum \ln(\hat{X}_A + 10^{-9})X_A \tag{1}$$

$$L_T = \sum \ln(\hat{X}_T + 10^{-9})X_T \tag{2}$$

$$L_{KL} = -\sum \frac{1}{2}(1 + \ln\sigma - \mu^2 - \sigma) \tag{3}$$

$$Loss = w_A L_A + w_T L_T + w_{KL} L_{KL} \tag{4}$$

where the weights of the three loss components were $w_A = \alpha / \ln(dim(X_A))$, $w_T = (1 − \alpha) / \ln(dim(X_T))$, and $w_{KL} = \beta / dim(Z)$. We adopted 0.1 and 0.015 for $\alpha$ and $\beta$, respectively (Supplementary Note 3). The VAE was trained with early stopping to reduce the training time and avoid overfitting. We used the RPH-kmeans[31] algorithm with random projection hashing to group the co-barcoded reads using their latent embeddings obtained from $\mu$.

### Weighted sampling for VAE training

We designed weighted sampling to balance the training set from microbes with different abundances. The *k*-mer histogram ($X_A$) is the combination of two Poisson distributions; one represents the erroneous *k*-mers (always with a low frequency)[46], and the other represents the true *k*-mer abundances[42]. If the co-barcoded reads of a specific barcode are from a low-abundance microbe, the Poisson distribution of the true *k*-mer abundance would also have its peak value obtained at low frequency. Therefore, the peak of the two Poisson distributions will be stacked together and result in a larger $max(X_A)$ (validated in Supplementary Fig. 5). Considering that the highest value of $X_A$ is negatively related to the abundance of the co-barcoded reads, we used a heuristic function $max(X_A)^2$ as the sampling weight for the barcode of the co-barcoded reads. The square was to make the low-abundance microbes have a much higher sampling weight. The calculated sampling weights were automatically used by the WeightedRandomSampler of PyTorch to construct a balanced training dataset.

### Multi-thresholding reassembly for low-abundance microbes

For co-barcoded read binning, we assembled reads in each cluster independently using MEGAHIT (v1.2.9)[30]. We designed a multi-thresholding reassembly strategy (Fig. 1b) to improve the assembly qualities of low-abundance microbes by recollecting the reads from the low-abundance microbes that were misclustered into different bins using read depth thresholds. To calculate the read depth of the contigs assembled from each read cluster (denoted as contigs$_{bin}$), we aligned the input reads to contigs$_{bin}$ using BWA-MEM (v0.7.17)[41] and calculated

the read depth for each contig using the "jgi_summarize_bam_contig_depths" program in MetaBat2 (v2.12.1)[47]. To collect the reads of low-abundance microbes, we extracted the reads that could not be mapped to the high-depth contigs (with read depth > $t_i$) in contigs$_{bin}$ and assembled them using the standard short-read assembler, metaSPAdes (v3.15.3)[29]. This step can be substituted by MEGAHIT to reduce the running time, and can also integrate available contigs to guide the path resolution in metaSPAdes to make the assembly of low-abundance microbes more efficient (e.g., using the contigs assembled by metaSPAdes from the whole read dataset as the input to "--untrusted-contigs" of metaSPAdes, which is used in our experiments and optional for Pangaea). We repeated this procedure with a range of thresholds ($T = \{t_i | i = 1, 2, ...\}$) producing contigs$_{low}$. We chose $T = \{10, 30\}$ for all the experiments, which worked well for both the low- and the high-complexity microbial communities.

### Ensemble assembly
We use ensemble assembly to avoid incomplete metagenome assembly caused by the mis-binning of previous modules. For linked-read assembly, the ensemble strategy includes two steps: (i) we use contigs$_{bin}$ (contigs assembled from each read bin), contigs$_{low}$ (contigs from multi-thresholding reassembly), contigs$_{local}$ (contigs from the local assembly of Athena) and contigs$_{ori}$ (contigs assembled from short-reads by metaSPAdes [3.15.3]) using an OLC-based assembler, metaFlye (v2.8) with the "--subassemblies" parameter; (ii) quickmerge (v0.3)[48] was used to merge the contigs from step (i) and Athena contigs. For the assembly of short-reads with virtual barcodes, we substituted the metaSPAdes in step (i) and Athena in step (ii) with the corresponding hybrid assemblies (contigs generated from hybrid-SPAdes [3.15.3][16] or OPERA-MS [v0.8.3][15]). Step (ii) is optional for linked-read assembly since Athena is already integrated in the Step (i).

### Detecting circular contigs
We adopted the circularization module of Lathe[11] to detect circular contigs in all the assemblies. The module needs long-reads as input which is not available for linked-read assembly, so we modified the alignment and assembly parameters in the circularization module to accept contigs as input, and merged the contigs$_{ori}$, contigs$_{bin}$, contigs$_{low}$, and contigs$_{local}$ as "pseudo long-reads" for running it.

### Reconstructing physical long fragments based on reference genomes
We reconstructed the physical long fragments from linked-reads of ATCC-MSA-1003 to calculate $N_{F/B}$. The linked-reads were mapped to the reference genomes using BWA-MEM (v0.7.17)[41] with option "-C" to retain the barcode information in the alignment file, followed by sorting based on read alignment coordinates using SAMtools (v1.9)[49]. We connected the co-barcoded reads into long fragments if their coordinates were within 10 kb on the reference genome. Each fragment was required to include at least two read pairs and to be no shorter than 1 kb.

### Metagenome assembly of the other assemblers on different datasets
The 10x, stLFR, and TELL-Seq sequencing datasets were demultiplexed to generate raw linked-reads using Long Ranger (v2.2.0)[50], stLFR_read_demux (Git version 3ecaa6b)[17] and LRTK (Git version 28012df)[39], respectively. The linked-reads were assembled using metaSPAdes (v3.15.3)[29], MEGAHIT (v1.2.9)[30], cloudSPAdes (v3.12.0-dev)[29], Athena (v1.3)[14] and Supernova (v2.1.1)[20]. Supernova does not accept raw stLFR linked-reads as input because its barcode processing was hard-coded for 10x Genomics, so we applied stlfr2supernova_pipeline (https://github.com/BGI-Qingdao/stlfr2supernova_pipeline; Git version 95f0848) to convert the barcode format of stLFR. The scaffolds produced by Supernova were broken into contigs at successive "N"s longer than 10 before evaluation. The datasets with both short- and

long-reads were assembled using MetaPlatanus (v1.3.1)[22], OPERA-MS (v0.8.3)[15], hybridSPAdes (v3.15.3)[16], metaSPAdes (v3.15.3, only short-reads) and Athena (v1.3, short-reads with virtual barcodes). The PacBio CLR long-reads from S1 and S2 were assembled using metaFlye (v2.8)[51] with the "--pacbio-raw" parameter. All the assemblers were run with default parameters. For measuring computational performance, we set the threads of all the assemblers to 100 (if the assembler had this option), and used the command "/usr/bin/time -v" to report the system time, user time, and maximum RSS consumed by the programs.

### Benchmarking on the mock microbial communities
The reference genomes of ATCC-MSA-1003 and ZYMO were downloaded from the NCBI reference databases (Supplementary Table 1) and the previous study on ZYMO[33], respectively. Reference genomes of CAMI-high were from the CAMI challenge website (https://data.cami-challenge.org/participate). The contigs assembled from the three mock communities were assessed using MetaQUAST (v5.0.2)[52], with the option "-m 1000 --fragmented --min-alignment 500 --unique-mapping" to enable the alignment of fragmented reference genomes and discard ambiguous alignments. The p-values of differences in the NA50, NGA50, and genome fractions of different assemblers were obtained using the Wilcoxon signed-rank test performed by the wilcox.test function under package stats (v4.4.0) of R. This function was run with "paired = TRUE", "exact = TRUE", "conf.int = TRUE" to pair the two groups tested, and get the p-value and confidence interval. The effective size statistic was calculated by $Z / \sqrt{N}$, where $Z$ was the Z-score obtained from the Wilcoxon Rank Sum test, and N was the total number of observations across both groups. NGA50[53] is a balanced evaluation metric considering the assembly length for the strain, the misassemblies, and the contig continuity of the assembly on the strain.

### Contig binning and MAG quality evaluation
We aligned the linked-reads (or short-reads) to the contigs using BWA-MEM (v0.7.17)[41] and calculated the read depths using "jgi_summarize_bam_contig_depths" in MetaBat2 (v2.12.1)[47]. The contigs with read depths were binned into MAGs using MetaBat2 (v2.12.1) with default parameters. VAMB binning was performed by VAMB (v3.0.3)[36] with default parameters. CheckM (v1.1.2)[54] was used to report the completeness and contamination of the MAGs. ARAGORN (v1.2.38)[55] and barrnap (v0.9)[56] were used to annotate the transfer RNAs (tRNAs) and rRNAs (5S, 16S, and 23S rRNAs), respectively. According to standard criteria of the minimum information about MAGs[57], we classified the MAGs into near-complete (completeness > 90%, contamination < 5%, and could be detected 5S, 16S, and 23S rRNAs, and at least 18 tRNAs), high-quality (completeness > 90%, and contamination < 5%), medium-quality (completeness ≥ 50%, and contamination < 10%), and low-quality (the other MAGs).

### Annotation of the MAGs and the closest reference genomes
The contigs were annotated using Kraken2 (v2.1.2) with the custom database built from the NT database of NCBI (Aug 20, 2022). We used the "--fast-build" option of kraken2-build to reduce the database construction time. Subsequently, the "assign_species.py" script from the https://github.com/elimoss/metagenomics_workflows[11,14] was used to annotate MAGs as species (if the fraction of contigs belonging to the species was more than 60%) or genus (otherwise) based on contig annotations. The closest reference genomes of the NCMAGs that can be annotated at species-level were identified using GTDB-Tk (v2.1.0; database version r207)[58], which also reported the alignment identities and alignment fractions between them.

### Statistics and reproducibility
No statistical method was used to predetermine sample size. The sample size for human participants (three human participants, referred to as S1, S2, and S3) was chosen considering the previous study[14],

where two human participants were enrolled. No data were excluded from the analyses. The three participants were randomly chosen from the staffs in Kangmeihuada GeneTech Co., Ltd. No different treatments were given to different participants during experiments and outcome assessment. The Investigators were not blinded to allocation during experiments and outcome assessment.

## Reporting summary

Further information on research design is available in the Nature Portfolio Reporting Summary linked to this article.

## Data availability

The 10x Genomics linked-reads of the ATCC-MSA-1003 mock community used in this study are available in the NCBI SRA database under accession code SRR12283286. The stLFR and TELL-Seq sequencing data of ATCC-MSA-1003 generated in this study have been deposited in the NCBI SRA database under accession code SRR21422848 and SRR21422847, respectively. The stLFR sequencing data of the three human gut microbiomes (S1, S2, and S3) generated in this study have been deposited in the NCBI SRA database under accession code SRR28959570, SRR28959569, and SRR28959571, respectively. The MAGs generated by Pangaea from S1, S2, and S3 in this study have been deposited in the European Nucleotide Archive (ENA) project under accession code PRJEB65432. The PacBio CLR long-reads of ATCC-MSA-1003, S1, and S2 used in this study are available in the NCBI SRA database under accession code SRR12371719, SRR19505636, and SRR19505632, respectively. The ONT long-reads of ZYMO used in this study are available in the NCBI SRA database under accession code ERR3152366. Source data are provided with this paper.

## Code availability

Codes of Pangaea are available at https://github.com/ericcombiolab/Pangaea[59].

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

## Acknowledgements

The design of the study and the collection, analysis, and interpretation of the data were partially supported by the Young Collaborative Research Grant (C2004-23Y, L.Z.), HMRF (11221026, L.Z.), the Science Technology and Innovation Committee of Shenzhen Municipality, China (SGDX20190919142801722, XD.F.), the open project of BGI-Shenzhen, Shenzhen 518000, China (BGIRSZ20220012, L.Z. and BGIRSZ20220014, K.J.Y.), the Hong Kong Research Grant Council Early Career Scheme (HKBU 22201419, L.Z.), HKBU Start-up Grant Tier 2 (RC-SGT2/19-20/SCI/007, L.Z.), HKBU IRCMS (No. IRCMS/19-20/D02, L.Z.). We thank Tom Chen and Yong Wang from Universal Sequencing Technology for providing the TELL-Seq sequencing data of the ATCC-MSA-1003 mock community. We thank Arend Sidow for his comments to improve the manuscript's language and structure. We also thank the Research Committee of Hong Kong Baptist University and the Interdisciplinary Research Clusters Matching Scheme for their kind support of this project.

## Author contributions

L.Z. conceived the study. Z.M.Z., L.Z., and J.X. designed the Pangaea algorithms. Z.M.Z. and J.X. implemented the Pangaea software. L.Z. and Z.M.Z. conceived the experiments. Z.M.Z. and J.X. conducted the experiments. Z.M.Z., L.Z., and J.X. analyzed the results. H.B.W. drew and analyzed the circos plots. C.Y. generated and analyzed the statistics of the three types of linked-reads. Z.M.Z. and L.Z. wrote the manuscript. XD.F., Y.F.H., Z.Y., Y.C., and LJ.H. sequenced the stLFR linked-reads. A.P.L. and K.J.Y. revised the paper and supported the project. All authors reviewed the manuscript.

## Competing interests

L.J.H. is an employee of Kangmeihuada GeneTech Co., Ltd (KMHD). The remaining authors declare no competing interests.
