## [Peer Review File · Nature Communications]

Exploring high-quality microbial genomes by assembling short-reads with long-range connectivityREVIEWER COMMENTS

Reviewer #1 (Remarks to the Author):

Utilizing short reads with physical barcodes and employing a combination of short and long reads for microbial genome assembly is computationally challenging. In this work, Zhang et al. developed a computational tool, Pangaea, to address this issue. Pangaea leverages a deep learning-based binning algorithm and employs more targeted thresholds for varying abundances to enhance assembly performance, ultimately leading to improved assembly quality.

This is an interesting work. However, we have several concerns about the current manuscript. We are quite open to reading the revised version if the authors can address those concerns in a satisfactory manner.

1. The datasets used for tool development and validation heavily rely on MOCK MSA 1003. Although real microbial communities (e.g., three stool samples) were incorporated, the absence of ground truth in real samples makes it challenging to comprehensively evaluate the assembly results. The simulated sequencing datasets, which should primarily serve for performance evaluation, appear to be limited.

2. When compared with previous tools, the choice of software lacks convincing justification.

3. The benchmarking criteria seem somewhat narrow, with focusing more on assembly length and not providing a comprehensive evaluation of assembly accuracy.

4. Line 61, regarding the statement “Several assemblers have been developed for assembling linked-reads...”, does the authors specifically refer to linked-reads assemblers for metagenomes? Currently, there are dozens of available assemblers for general linked-reads (see: <https://github.com/pontushojer/awesome-linked-reads>). The authors need to

clarify the distinction between linked-reads assemblers for metagenomes and other linked-reads assemblers. Furthermore, while it's understood that comparing with all existing software is impractical, there indeed needs to be a stronger rationale provided to justify the reason for selecting the tools mentioned (on line 107) in this manuscript. Otherwise, it raises concerns about potential bias.

5. In the comparison results, cloudSPAdes' data seems to be absent in several places (e.g., missing in Fig. 2, 3, 4, and Table S6, etc.) due to its high memory and hardware requirements. It's inappropriate for the authors to claim they've made a comparison with cloudSPAdes yet fail to present the results consistently.

6. Line 160: the authors stated "there were no available binning tools for linked-reads," leading them to connect the co-barcoded reads using a single "N" into long-reads to "enable these long-read binning tools on linked-reads." We wonder if it is possible to employ regular assemblers to handle barcoded reads separately, rather than resorting to a somewhat forced use of long-read-based binning tools?

7. Line 186: Given that the known MOCK MSA 1003 contains only 20 species, the author chose 15 as the bin number. How was the decision made to use 30 as the bin number for human gut microbiomes? And how would one determine the appropriate bin number for a new microbiome sample?

8. Line 192: Is this value of T generally applicable to all samples, or is it specific to MOCK MSA 1003? In this context, does the role of MSA 1003 encompass both the exploration of the T value and the evaluation of the final performance?

9. Line 200: in order to evaluate the barcode specificity on assembly, the authors compared "The contigs of Pangaea from stLFR and TELL-Seq linked-reads with the contigs of Athena and Supernova from 10x linked-reads". This comparison introduces two variables - different software and different data types. As a result, the conclusion doesn't directly correlate with the authors' statement that "Barcode specificity is critical for linked-read assembly". Why wasn't the comparison conducted in a more comprehensive manner, for

instance, by using all three software tools with each of the three data types and then comparing the results?

10. Line 227: The authors simulated 143.04Gb stLFR linked reads for the Zymo mock community. Given that simulated sequencing data comes with ground truth, it's suitable for evaluating the performance among different methods. Moreover, simulation data isn't restricted by the limitations of MOCK samples, e.g., the limited number of species and availability. Thus, the use of simulation data allows for a more systematic and direct comparison of software performance. However, the simulation data (for ZYMO) has a high redundancy with MOCK MSA 1002, both possessing a very simple microbial composition (less than 20 species). It would be beneficial to simulate more complex sequencing data that mirrors real microbial community samples (perhaps encompassing around 200 species) to demonstrate the software's capability in handling real microbiome samples.

11. One limitation of the current study is that the evaluation framework is predominantly built upon length metrics like N50. Although the authors assessed the “collinearities between NCMAGs and their closest reference genomes” on line 302, there is still a noticeable absence of a systematic and effective evaluation of assembly quality. For instance, assessments of completeness, redundancy (using tools like CheckM), assembly errors (using tools like QUAST or MetaQuast), chimeric assemblies (using tools like GTDB-TK), and contiguity (metrics like N70, and N90) should be considered. Clearly, for real microbial community samples, such evaluations are challenging, but for MOCK communities, the above assessments are feasible. Additionally, to systematically evaluate assembly accuracy, the inclusion of more simulated sequencing data is indispensable.

12. Line 352: A similar concern arises when considering the hybrid assembler in the comparison/benchmarking. There are several other notable options like nf-core/mag (by Sabrina Krakau et al.) and MetaPlatanus (by Rei Kajitani et al.), among others. While we might regard OPERA-MS as the best hybrid assembler as of 2019, the two software options mentioned above were published subsequently. If these are not included in the comparison, the authors need to provide a very compelling rationale.

Reviewer #2 (Remarks to the Author):

Zhang et al. presents a method (Pangea) that takes as input metagenome co-barcoded reads and outputs assemblies and MAGs. The binning is used to connect co-barcoded reads that were not connected initially. Additionally, the method uses a technique called multi-thresholding to improve binning of low abundance samples.

In general, this is an interesting paper where the authors propose that the addition of the co-barcoding and initial binning of the co-barcoded reads improve assembly and binning from metagenomics datasets. This sounds very likely, however more comparisons are needed to fully support the claim.

Major points:

1. Quite a lot of data is generated for each sample, much more than in a “traditional” gut microbiome sample approach. I.e. the Zymo/ATCC datasets are between 100-150Gb of data and includes on 20 strains. For instance, for the ATCC Mock dataset with 132Gb of data, the coverage of strains will be:

0.02% = 26 Mb

0.18% = 240 Mb

1.8% = 2.4 Gb

18% = 24 Gb

This means that at the lowest abundance (0.02%) there might still be 5-10X of data available, e.g. 26Mb and genomes being between 2-5Mb. At 0.18% there is between 50-100X of data available, etc. This is quite a lot of data and more than what one would normally generate for e.g. gut microbiome samples (which they compare to).

Similarly, the three gut microbiome samples consist of 136, 132 and 51 Gb of data!

2. Related to the above, traditional de novo assembly algorithms break at when genomes are at high depth, downsampling would be interesting as this may improve contiguity (e.g. 5Gb, 10Gb, 20Gb or similar).

3. As Pangea also include binning compared to traditional assemblers it does not make sense to compare how many NCMAGs are generated from Pangea compared to tools that only do assembly (Megahit, Spades). It would, however, be interesting how state-of-the-art assemblers and binners such as VAMB, SemiBin2 or other methods perform on the same samples, e.g. short-read+binners and long-read+binners. My guess is that they will be able to assemble and bin the data quite well. If Pangea outperform the methods it would be very interesting.

4. Ln 441-443: “In our experiments, we found that long-read assemblies had 60.98% fewer NCMAGs than linked-read assemblies from Pangaea (Supplementary Note 4), indicating that some microbes might be lost due to insufficient long-read sequencing depth.”

I don't understand why PacBio and ONT on the Zymo and ATCC datasets are downsampled? They only represent 0.5-2% of Gbp compared to the linked short reads. Please compare to when using a realistic throughput of data from these technologies. Because of this, currently, the results sections comparing Pangea assembly statistics to long-read based data does not really make sense.

For instance, ATCC-MSA-1003 has been used for assembly and binning of PacBio reads in Benoit et al., where assembly and binning using metaMDBG generate circularized genomes and almost complete genomes (Benoit et al., biorxiv, 2023). Likewise, ONT data has been used to generate many NCMAGs from various datasets.

5. Costs is mentioned several times as an advantage of the barcoded reads compared to long read, however no information on costs is given. What are the costs of generating the data linked data compared to short-read and long-read?

6. First paragraph in the Results section “Workflow of Pangea” is a bit difficult to follow. This makes it difficult to intuitively understand how the method works. Please consider re-writing.

7. Ln 190-199: I am not convinced that Thresholding improves the binning. There is no statistical test done and inspecting Supplementary Table 5 is a mix of blue and red (ie. better/worse). For many the differences are also very small in the number of bp. Maybe a statistical test could show whether it is better.

8. Ln 162-163: “In order to enable these long-read binning tools on linked-reads, the co-barcoded reads were connected using a single “N” into long-reads”. For the comparison to long read binners, was the order of the linked reads known? Connecting the co-barcoded short reads with N in a random fashion is likely going to be difficult for the binners.

Minor points:

9. Please reference Nissen et al., as they are the first to introduce VAEs for binning which is used here (disclaimer: I am a co-author of that paper).

10. Ln 134-135: “We observed there was a negative correlation between $\max(XA)$ and the abundance of the co-barcoded reads (Supplementary Figure 1).” I don’t understand why this is a negative correlation, but perhaps it is because I am not fully sure what $\max(Xa)$ is? Also I don’t understand how that can be seen from Supplementary Figure 1.

11. Ln 137: Describe RPH-kmeans, what is RPH?

12. Ln 141-142: “This module aligns short-reads to contigs and removes them if they are from the contigs with abundances above certain thresholds”. Please elaborate a bit more.

13. Ln 171-187: Comparison between ASM_b and ASM_notB. Can you report precision/recall instead of N50 and NGA50? Ie what was the precision/recall of a genome reconstruction between the two approaches?

14. Ln 180: What is bin number, I assume this is number of clusters? This has not been introduced before. Do you have to set a number beforehand? How is this number set?

15. Figure 3.1. Please add meaningful label to y-axis (currently now is "value")

Response to reviewer Comments

Reviewer #1 (Remarks to the Author):

Utilizing short reads with physical barcodes and employing a combination of short and long reads for microbial genome assembly is computationally challenging. In this work, Zhang et al. developed a computational tool, Pangaea, to address this issue. Pangaea leverages a deep learning-based binning algorithm and employs more targeted thresholds for varying abundances to enhance assembly performance, ultimately leading to improved assembly quality.

This is an interesting work. However, we have several concerns about the current manuscript. We are quite open to reading the revised version if the authors can address those concerns in a satisfactory manner.

1. The datasets used for tool development and validation heavily rely on MOCK MSA 1003. Although real microbial communities (e.g., three stool samples) were incorporated, the absence of ground truth in real samples makes it challenging to comprehensively evaluate the assembly results. The simulated sequencing datasets, which should primarily serve for performance evaluation, appear to be limited.

Response: Thank you for your insightful comments. In the revised manuscript, we simulated 5G, 10G, 20G, 50G and 100G linked-reads for CAMI-high (a high-complexity community containing 596 genomes) and together with ATCC-MSA-1003 (a low-complexity community) to evaluate the performance of Pangaea (Table 1 and 3; Supplementary Table 4, 6 and 14).

The manuscript has been revised as:

1. Line 168-170 (for co-barcoded read binning):

We observed ASM_B generated one more NCMAg than ASM_{-B} for CAMI-high ($ASM_B = 14$, $ASM_{-B} = 13$; Supplementary Table 4).

2. Line 199-201 (for multi-thresholding reassembly):

The assembly with multi-thresholding reassembly generated significantly higher genome fractions for the 596 low-abundance microbes than the assembly without this module (Wilcoxon paired rank-sum test p-value = $1.99e-05$; Supplementary Table 6).

3. Line 236-244: (compare with the other assemblers)

For simulated linked-reads from CAMI-high, Pangaea generated the highest total assembly length, genome fraction, N50 (1.87 times on average), overall NA50 (1.61 times on average), NA50 per strain (1.44 times on average), and NGA50 per strain (1.58 times on average) than the other assemblers (Table 1). Although Pangaea and Athena got comparable overall NA50s (Pangaea produced more sequences), the NGA50 per strain of Pangaea was much higher than that of Athena (Pangaea = 54.41Kb, Athena = 49.90Kb; Table 1). Pangaea generated the largest number

of genomes for which the assemblies covered at least 50% (non-zero NGA50s; Pangaea = 195, Athena = 180, Supernova = 175, cloudSPAdes = 177, metaSPAdes = 177, MEGAHIT = 180; Table 1). These results indicate Pangaea performed well on both the high-complexity dataset and the low-abundance microbes.

4. Line 353-358 (investigating the impact of sequencing depth):

Pangaea achieved better NA50 values than the other tools except for Athena on CAMI-high. On datasets under 50Gb from CAMI-high, Athena achieved better overall NA50s; however, this higher contiguity came at the cost of significantly shorter total assembly lengths (Supplementary Table 14). On the dataset of 100Gb from CAMI-high, Pangaea achieved the highest overall NA50, NGA50 per strain and the number of non-zero NGA50s (Supplementary Table 14).

5. Line 382-385 (with virtual barcodes):

We compared Pangaea-MetaPlatanus, Pangaea-Hybridspades and Pangaea-Operams to their original assemblers as well as Athena and metaSPAdes, and observed that virtual barcodes could prominently increase contig N50, overall NA50 and average NGA50 per strain on ATCC-MSA-1003 and CAMI-high (Table 3; Figure 6 a).

2. When compared with previous tools, the choice of software lacks convincing justification.

Response:

- For short-read assemblers, metaSPAdes and MEGAHIT are the two best performing short-read assemblers based on the previous studies (Zhang et al. 2023; Vollmers, Wiegand, and Kaster 2017; Van der Walt et al. 2017).
- For linked-read assemblers, we benchmarked Athena, Supernova, and cloudSPAdes, which were designed for metagenome assembly from <https://github.com/pontushojer/awesome-linked-reads>.
- Below are the descriptions of the tools from <https://github.com/pontushojer/awesome-linked-reads> with “assembly” in the “Category” column.

Names	Descriptions
Aquila	A tool for structural variation call.
Aquila_stLFR	A tool for structural variation call.
Ariadne	A tool for linked-reads deconvolution.
arcs	A tool for contig scaffolding.
Athena	A tool for metagenome assembly (Benchmarked in the manuscript).
cloudSPAdes	A tool for metagenome assembly (Benchmarked in the manuscript).
HAST	A tool for human genome assembly. The input needs paternal and maternal genomes.
MetaTrass	A tool for reference-based metagenome assembly. We have discussed this tool in the Introduction.
Minerva	A tool for linked-reads deconvolution.

MTG-Link	A tool for post-assembly gap closer.
NPGREAT	A tool to assemble human subtelomere regions.
Pangaea	A tool for metagenome assembly (This is our tool).
Pyslr	A tool to improve contig scaffolding.
QuickDeconvolution	A tool for linked-reads deconvolution.
Scaff10x ($\leq v4.1$)	A tool for contig scaffolding.
SpLitter (alt)	A tool to improve HiFi assembly using synthetic long-reads.
stLFRdenovo	A tool to accommodate Supernova for stLFR linked-reads (The performance is same as Supernova).
Supernova	10x Genomics official assembler (Benchmarked in the manuscript).
Tigmint	A tool for contig error correction.

- For hybrid assembly, we included three tools (OPERA-MS, hybridSPAdes and MetaPlatanus) designed specifically for metagenome hybrid assembly.

3. The benchmarking criteria seem somewhat narrow, with focusing more on assembly length and not providing a comprehensive evaluation of assembly accuracy.

Response: In the revised manuscript, we have evaluated the assembly using contig N50, N70, N90, NA50 (the N50 after breaking contigs at misassemblies), NGA50 per strain (the NA50 using the reference genome length as total length instead of assembly length), and the number of NCMAGs (considering MAG completeness, contamination, and annotations of RNAs), misassemblies (including chimeric assembly) from MetaQUAST and sequencing identity (Tables 1-3; Supplementary Table 12 and 16).

4. Line 61, regarding the statement “Several assemblers have been developed for assembling linked-reads...”, does the authors specifically refer to linked-reads assemblers for metagenomes? Currently, there are dozens of available assemblers for general linked-reads (see: <https://github.com/pontushojer/awesome-linked-reads>). The authors need to clarify the distinction between linked-reads assemblers for metagenomes and other linked-reads assemblers. Furthermore, while it's understood that comparing with all existing software is impractical, there indeed needs to be a stronger rationale provided to justify the reason for selecting the tools mentioned (on line 107) in this manuscript. Otherwise, it raises concerns about potential bias.

Response: We have investigated all tools with the “assembly” tag in the “Category” column listed at <https://github.com/pontushojer/awesome-linked-reads>. All the genome assembly tools that could be applied in metagenome assembly have been included in our experiments. Please see the table in question 2 for more detailed interpretations.

5. In the comparison results, cloudSPAdes' data seems to be absent in several places (e.g., missing in Fig. 2, 3, 4, and Table S6, etc.) due to its high memory and hardware requirements.

It's inappropriate for the authors to claim they've made a comparison with cloudSPAdes yet fail to present the results consistently.

Response: We found cloudSPAdes was sensitive to reads with low quality. It requires high memory and hardware requirements to deal with these reads. We tried to remove these reads using fastp and cloudSPAdes can generate output for all cases, except 10x Genomics and stLFR linked-read datasets of ATCC-MSA-1003. Both two datasets required more than 2TB memory, which exceeded our hardware configuration. For the other datasets, we have updated the results by incorporating assembly statistics of cloudSPAdes and revised the manuscript as:

1. Line 259-262:

Moreover, Pangaea achieved substantially higher N50s than all the other assemblers for both S1 (1.44 times of Athena, 1.06 times of Supernova, 3.65 times of cloudSPAdes, 4.71 times of MEGAHIT, 4.50 times of metaSPAdes; Table 2) and S2 (1.57 times of Athena, 2.58 times of Supernova, 6.33 times of cloudSPAdes, 8.00 times of MEGAHIT, 7.99 times of metaSPAdes; Table 2).

2. Line 269-271:

Pangaea generated NCMAGs (Figure 3 a, e and i) of 24, 17 and 9 for S1, S2 and S3, which were much more than those generated by Athena, Supernova, cloudSPAdes, MEGAHIT and metaSPAdes.

3. Line 303-309:

The NCMAGs generated by different assemblers and their closest reference genomes had comparable average alignment identities (Pangaea: 98.16%, Athena: 98.12%, Supernova: 98.31%, cloudSPAdes: 98.77%, MEGAHIT: 98.7%, metaSPAdes: 98.8%) and average alignment fractions (Pangaea: 87.9%, Athena: 88.6%, Supernova: 88.4%, cloudSPAdes: 88.5%, MEGAHIT: 88%, metaSPAdes: 90%; Supplementary Table 13), while Pangaea produced significantly more species-level NCMAGs than the other assemblers (Pangaea: 29, Athena: 21, Supernova: 14, cloudSPAdes: 2, MEGAHIT: 1, metaSPAdes: 1; Supplementary Table 13).

4. Line 335-338

cloudSPAdes obtained a high-quality MAG for *B. adolescentis*, but this MAG could not be annotated with any rRNAs, and the contig continuity was much lower than the corresponding MAG from Pangaea (Pangaea = 2.17Mb, cloudSPAdes = 329.67Kb).

6. Line 160: the authors stated "there were no available binning tools for linked-reads," leading them to connect the co-barcoded reads using a single "N" into long-reads to "enable these long-read binning tools on linked-reads." We wonder if it is possible to employ regular assemblers to handle barcoded reads separately, rather than resorting to a somewhat forced use of long-read-based binning tools?

Response: Pangaea aims to perform metagenome assembly on linked-reads from 10x Genomics, stLFR and TELL-Seq platforms, where co-barcoded linked-reads could only have shallow coverage (0.3X-0.4X; Supplementary Note 1) of the target long fragments. Thus, assembly on co-barcoded reads is not feasible for such cases.

As the long-read binners will skip k-mers containing N (see code in METABCC-LR; the same for LRBinner), the process of connecting co-barcoded reads using “N”s will not affect the performance of those long-read binners. As this part is not directly related to metagenome assembly, we moved this part to Supplementary Note 2.

7. Line 186: Given that the known MOCK MSA 1003 contains only 20 species, the author chose 15 as the bin number. How was the decision made to use 30 as the bin number for human gut microbiomes? And how would one determine the appropriate bin number for a new microbiome sample?

Response: The bin number is a hyperparameter of Pangaea, which is positively related to the Shannon diversity of the dataset (α *Shannon diversity).

It is a trade-off for bin number selection: choosing a low bin number might result in limited power to deconvolve dataset complexity, and a high bin number might lead to poor assembly due to assigning reads of the same microbe into different bins. We manually set 30 for the three human gut microbiomes (they have close Shannon diversity), which could generate the best assembly. This enables us to calculate $\alpha=8$ and bin numbers can be calculated as 8 *Shannon diversity. Thus, we can calculate the number of bins for ATCC-MSA-1003 as 15 instead of 20. Also, we observed the assemblies were not sensitive to bin number if the bin number was not shifted far away from 8 *Shannon diversity (Supplementary Figure 3). We added a paragraph to describe the process of determining the bin number:

Line 174-183:

We observed the number of read bins (k) for read binning was positively correlated with the diversities of metagenome sequencing datasets ($k = \alpha * \text{Shannon Diversity}$; Supplementary Table 2; Methods). To determine the coefficient α , we chose a k that worked well on all three human gut microbiomes ($k = 30$) and calculated the coefficient α as 8 by linear regression. Our results showed that k could influence both the precision and recall of read binning (a large k resulted in high binning precision and low recall; Supplementary Figure 2). The k is a trade-off between generating read bins with low complexities (large k) or keeping more reads from the same microbes in the same bin (small k). Although k would influence read binning performance, the assembly results seemed robust if k was not shifted too much from the value calculated from the formula ($k = 15$) for ATCC-MSA-1003 (e.g., assemblies from $k = 10, 15, 20$ were comparable; Supplementary Figure 3).

8. Line 192: Is this value of T generally applicable to all samples, or is it specific to MOCK MSA 1003? In this context, does the role of MSA 1003 encompass both the exploration of the T value and the evaluation of the final performance?

Response: T was set independently of the datasets. We used T to improve the assemblies of low-abundance microbes, which typically refers to the contig with low depth. We aligned the reads to contigs and only selected the reads if they originated from contigs with a depth below 10x or 30x.

We modified the first paragraph of the section “Multi-thresholding reassembly improves assemblies of low-abundance microbes” to describe the choosing of our T:

Line 186-190:

Pangaea improves assemblies of low-abundance microbes by gradually removing the reads from high-abundance microbes from the assembly graph with multiple abundance thresholds (represented by T). As this module aims to improve low-abundance microbial assembly, we only consider reads from contigs with average depths lower than 10x (ultra-low) and 30x (low). The two thresholds have been validated by ATCC-MSA-1003 (Supplementary Note 3).

9. Line 200: in order to evaluate the barcode specificity on assembly, the authors compared “The contigs of Pangaea from stLFR and TELL-Seq linked-reads with the contigs of Athena and Supernova from 10x linked-reads”. This comparison introduces two variables - different software and different data types. As a result, the conclusion doesn't directly correlate with the authors' statement that “Barcode specificity is critical for linked-read assembly”. Why wasn't the comparison conducted in a more comprehensive manner, for instance, by using all three software tools with each of the three data types and then comparing the results?

Response: In this section, we aim to investigate the impact of barcode specificity on the performance of Pangaea. Because barcode specificity refers to different platforms (10x Genomics, stLFR and TELL-Seq), we modified this section to compare the performance of Pangea on 10x Genomics, stLFR and TELL-Seq linked-reads. The revised manuscript is shown below:

Line 202-216:

Barcode specificity is critical for linked-read assembly

We applied Pangaea to linked-reads from 10x Genomics, TELL-Seq and stLFR of ATCC-MSA-1003 to investigate the impact of barcode specificity on the performance of Pangaea (Supplementary Table 2; Methods). The linked-reads from stLFR and TELL-Seq yielded much lower $N_{F/B}$ (stLFR: 1.54, TELL-Seq: 4.26) compared to those obtained from 10x Genomics (10x Genomics: 16.61 Supplementary Note 1). The contigs from Pangaea on stLFR and TELL-Seq datasets had substantially higher N50s (1.44 times on average; Supplementary Table 7) and higher overall NA50s (1.43 times on average; Supplementary Table 7) than the assembly on 10x Genomics linked-reads. For those 15 strains with abundance > 0.1% (Supplementary Table 8), the assembly on stLFR linked-reads achieved significantly higher strain NA50s (p-value = 0.0353; Methods) and NGA50s (p-value = 0.0479; Methods) than those on 10x Genomics dataset. The same trend was also observed between the assemblies on TELL-Seq and 10x Genomics datasets (Supplementary Table 8). For the remaining 5 strains with abundances of 0.02%, all datasets cannot be assembled with high genome fractions, making it infeasible to compare their NGA50s (Supplementary Table 9). These results suggest that linked-reads with high barcode specificity could produce better metagenome assemblies using Pangaea.

We have also compared Pangaea, Athena and Supernova on 10x Genomics, stLFR and TELL-Seq in the section “Pangaea generated high-quality metagenome assemblies on mock and simulated linked-read datasets”:

Line 220-235:

For TELL-Seq of ATCC-MSA-1003 (Table 1; Figure 2 b), Pangaea achieved the highest N50 (1.36Mb) and overall NA50 (649.47Kb) when compared with the statistics achieved by Athena (N50: 466.50Kb; NA50: 361.57Kb), Supernova (N50: 102.76Kb; NA50: 97.31Kb), cloudSPAdes (N50: 127.42Kb; NA50: 118.16Kb), MEGAHIT (N50: 128.07Kb; NA50: 112.51Kb) and metaSPAdes (N50: 112.34Kb; NA50: 105.63Kb) (Figure 2 a and c). When considering those 15 strains with abundances > 0.1% (Supplementary Table 8), Pangaea still generated significantly higher strain NA50s (Figure 2 e) and NGA50s (Figure 2 h) than Athena (NA50: p-value = 1.22e-4; NGA50: p-value = 6.10e-5), Supernova (NA50: p-value = 3.05e-4; NGA50: p-value = 3.05e-4), cloudSPAdes (NA50: p-value = 6.10e-5; NGA50: p-value = 6.10e-5), MEGAHIT (NA50: p-value = 6.10e-5; NGA50: p-value = 6.10e-5) and metaSPAdes (NA50: p-value = 6.10e-5; NGA50: p-value = 6.10e-5). A comparable trend was observed on the assemblies of 10x Genomics and stLFR linked-reads (Table 1; Figure 2 d, g, f and i). For the 5 strains with the lowest abundance (0.02%) of ATCC-MSA-1003, the assemblies of Pangaea had much higher genome fractions than those of Athena (9.40 times on average) and Supernova (47.87 times on average) on all three platforms (Supplementary Table 9), suggesting more genomic sequences could be assembled by Pangaea for low-abundance microbes.

10. Line 227: The authors simulated 143.04Gb stLFR linked reads for the Zymo mock community. Given that simulated sequencing data comes with ground truth, it's suitable for evaluating the performance among different methods. Moreover, simulation data isn't restricted by the limitations of MOCK samples, e.g., the limited number of species and availability. Thus, the use of simulation data allows for a more systematic and direct comparison of software performance. However, the simulation data (for ZYMO) has a high redundancy with MOCK MSA 1002, both possessing a very simple microbial composition (less than 20 species). It would be beneficial to simulate more complex sequencing data that mirrors real microbial community samples (perhaps encompassing around 200 species) to demonstrate the software's capability in handling real microbiome samples.

Response: We have simulated 5G, 10G, 20G, 50G and 100G linked-reads for a high-complexity community (CAMI-high), containing 596 genomes. The detailed information can be found in our answer to the question 1.

11. One limitation of the current study is that the evaluation framework is predominantly built upon length metrics like N50. Although the authors assessed the "collinearities between NCMAGs and their closest reference genomes" on line 302, there is still a noticeable absence of a systematic and effective evaluation of assembly quality. For instance, assessments of completeness, redundancy (using tools like CheckM), assembly errors (using tools like QUAST or MetaQuast), chimeric assemblies (using tools like GTDB-TK), and contiguity (metrics like N70, and N90) should be considered. Clearly, for real microbial community samples, such evaluations are challenging, but for MOCK communities, the above assessments are feasible. Additionally, to systematically evaluate assembly accuracy, the inclusion of more simulated sequencing data is indispensable.

Response: In the revised manuscript, we evaluated the performance of assembly using contig N50, N70, N90, NA50 (the N50 after breaking contigs at misassemblies), NGA50 per strain (the NA50 using the reference genome length as total length instead of contig length), and the number of NC MAGs (considering MAG completeness, contamination, and annotations of RNAs), misassemblies (including chimeric assembly) from MetaQUAST and sequencing identity (Tables 1-3; Supplementary Table 12 and 16). We have proved Pangaea outperformed the existing tools for most of these statistics.

12. Line 352: A similar concern arises when considering the hybrid assembler in the comparison/benchmarking. There are several other notable options like nf-core/mag (by Sabrina Krakau et al.) and MetaPlatanus (by Rei Kajitani et al.), among others. While we might regard OPERA-MS as the best hybrid assembler as of 2019, the two software options mentioned above were published subsequently. If these are not included in the comparison, the authors need to provide a very compelling rationale.

Response: Nf-core/mag is a tool based on hybridSPAdes, which has been benchmarked in our manuscript. In the revised manuscript, we have involved MetaPlatanus in the evaluation, and observed Pangaea could also improve its performance. The modifications have been shown in “Pangaea generated high-quality assemblies on short-reads with virtual barcodes from long-reads”:

Line 382-401:

We compared Pangaea-MetaPlatanus, Pangaea-Hybridspades and Pangaea-Operams to their original assemblers as well as Athena and metaSPAdes, and observed that virtual barcodes could prominently increase contig N50, overall NA50 and average NGA50 per strain on ATCC-MSA-1003 and CAMI-high (Table 3; Figure 6 a). All assemblers could not assemble with high genome fractions (>50%) for most of the 5 strains with the lowest abundance (0.02%) in ATCC-MSA-1003. However, Pangaea could generate more sequences for the informative long contigs (>10Kb) than the corresponding hybrid assembly tools (Supplementary Table 15).

We evaluated the capability of Pangaea to detect the extremely low-abundance microbes in ZYMO. For the two strains with abundances > 1%, Pangaea-Hybridspades generated much higher NGA50 than Hybridspades (2.23 times); the performance of Pangaea-MetaPlatanus and Pangaea-Operams was comparable with their original tools; this is acceptable as ZYMO is a simple metagenome where the existing hybrid assemblers can already assemble well for those high-abundance microbes in this dataset (Supplementary Table 15). For the two strains with abundance between 0.1% and 1%, Pangaea-Hybridspades and hybridSPAdes had comparable NGA50s, while Pangaea-MetaPlatanus (1.18 times) and Pangaea-Operams (1.45 times) produced better average NGA50 than MetaPlatanus and OPERA-MS, respectively (Supplementary Table 15). For the other two strains with abundance < 0.1%, Pangaea-Operams still obtained substantially higher average NGA50 than Operams (2.37 times), and all of the three models from Pangaea generated more sequences for long contigs (>10Kb) than their original

tools, respectively (Supplementary Table 15). All assemblers produced low-quality assemblies for the strain with an abundance < 0.01% (Supplementary Table 15).

Reviewer #2 (Remarks to the Author):

Zhang et al. presents a method (Pangea) that takes as input metagenome co-barcoded reads and outputs assemblies and MAGs. The binning is used to connect co-barcoded reads that were not connected initially. Additionally, the method uses a technique called multi-thresholding to improve binning of low abundance samples.

In general, this is an interesting paper where the authors propose that the addition of the co-barcoding and initial binning of the co-barcoded reads improve assembly and binning from metagenomics datasets. This sounds very likely, however more comparisons are needed to fully support the claim.

Major points:

1. Quite a lot of data is generated for each sample, much more than in a “traditional” gut microbiome sample approach. I.e. the Zymo/ATCC datasets are between 100-150Gb of data and includes on 20 strains. For instance, for the ATCC Mock dataset with 132Gb of data, the coverage of strains will be:

0.02% = 26 Mb

0.18% = 240 Mb

1.8% = 2.4 Gb

18% = 24 Gb

This means that at the lowest abundance (0.02%) there might still be 5-10X of data available, e.g. 26Mb and genomes being between 2-5Mb. At 0.18% there is between 50-100X of data available, etc. This is quite a lot of data and more than what one would normally generate for e.g. gut microbiome samples (which they compare to).

Similarly, the three gut microbiome samples consist of 136, 132 and 51 Gb of data!

Response: We used a relatively large amount of linked-reads for metagenomic assembly due to two reasons: 1. Linked-read sequencing commonly has a much higher PCR duplication rate (e.g. duplication ratio for S1: 61.73%) than short-read sequencing (<5%), which is due to their special library preparation protocol. Hence, we need more sequencing amount to collect sufficient informative reads. 2. The assemblers require sufficient co-barcoded reads to guarantee each long fragment could be well covered, because the long fragment coverage is only 0.3X-0.4X. Our previous study has shown insufficient sequencing depth for long fragments could significantly influence the reconstruction of long-range connectedness of short-reads and reduce the qualities of MAGs. More detained investigation for the impact of sequencing depth on linked-read

metagenome assembly can be found in our paper published Microbiome (cite the paper). We also subsampled sequencing data and evaluated the performance of Pangaea (See the answer of question 2).

2. Related to the above, traditional de novo assembly algorithms break at when genomes are at high depth, downsampling would be interesting as this may improve contiguity (e.g. 5Gb, 10Gb, 20Gb or similar).

Response: Thank you for your comments. As you suggested, we downsampled the datasets to 5G, 10G, 20G, 50G and 100G for ATCC-MSA-1003 (TELL-Seq), CAMI-high (stLFR), and S1 (stLFR) and observed more reads could significantly improve the performance of all linked-read assemblers. The results have been included in the section “Investigate the impact of sequencing depth on assembly results”:

Lines 345-370:

We assessed Pangaea’s effectiveness on datasets with varying sequencing depths by generating subsets of linked-reads at 5Gb, 10Gb, 20Gb, 50Gb, and 100Gb from ATCC-MSA-1003 (TELL-Seq), CAMI-high (stLFR), and S1 (stLFR) (Methods). We then compared Pangaea’s assembly performance on these subsampled read sets against other assembly algorithms. Increasing the sequencing depth improved the assembly quality of metagenomes across all three microbial community datasets for all involved assemblers (Supplementary Table 14).

Pangaea demonstrated superior performance compared to the other assemblers across all subsampling datasets. Pangaea obtained better overall NA50 and NGA50 per strain than the second-best assembler on all subsampled datasets from ATCC-MSA-1003 (Supplementary Table 14). Pangaea achieved better NA50 values than the other tools except for Athena on CAMI-high. On datasets under 50Gb from CAMI-high, Athena achieved better overall NA50s; however, this higher contiguity came at the cost of significantly shorter total assembly lengths (Supplementary Table 14). On the dataset of 100Gb from CAMI-high, Pangaea achieved the highest overall NA50, NGA50 per strain and the number of non-zero NGA50s (Supplementary Table 14).

Pangaea consistently generated a higher N50 than Athena on all the subsampled datasets on S1. On the 5Gb and 10Gb datasets of S1, Supernova achieved the highest N50 values, but this was accompanied by the shortest assembled sequence lengths relative to the other assemblers (Supplementary Table 14). Pangaea significantly outperformed Athena on the 100Gb dataset of S1 with respect to the number of NCMAGs (Pangaea = 18, Athena = 10), while they showed similar performance on the other subsampled datasets. These findings indicate that for human gut microbiomes, increased sequencing depth improves Pangaea’s performance in producing NCMAGs. Nevertheless, it should be emphasized that there is a positive relationship between data volume and the performance of all assemblers in generating NCMAGs. This may be because of the high PCR duplication rate of linked-read sequencing technologies [17], where 61.73% of stLFR linked-reads in S1 were marked as duplication (Methods). Therefore, the sequencing amount of linked-reads used for metagenome assembly is commonly higher compared to short-reads to ensure sufficient informative reads are given.

3. As Pangea also include binning compared to traditional assemblers it does not make sense to compare how many NCMAGs are generated from Pangea compared to tools that only do assembly (Megahit, Spades). It would, however, be interesting how state-of-the-art assemblers and bidders such as VAMB, SemiBin2 or other methods perform on the same samples, e.g. short-read+binners and long-read+binners. My guess is that they will be able to assemble and bin the data quite well. If Pangea outperform the methods it would be very interesting.

Response: The co-barcoded read binning module in Pangea is designed to improve genome assembly, and the output of Pangea is still a collection of contigs. We are still required to perform contig binning and group them into MAGs. We attempted to compare the number of NCMAG of the four strategies: Pangea (linked-reads)+VAMB, metaFlye(long-reads)+VAMB, metaSPAdes(short-reads)+VAMB and MEGAHIT(short-reads)+VAMB on S1 and S2. NCMAG is defined as MAGs with completeness > 90%, contamination < 5%, and with at least 18 tRNAs, one 5s rRNAs, one 16s rRNAs, and one 23s rRNAs. Pangea+VAMB (NCMAG: S1 = 21, S2 = 21) still generated the highest number of NCMAGs compared to metaFlye+VAMB (NCMAG: S1 = 6, S2 = 6), metaSPAdes+VAMB (NCMAG: S1 = 0, S2 = 0), and MEGAHIT+VAMB (NCMAG: S1 = 1, S2 = 0). This information has been added to the last paragraph of Discussion:

Line 476-480:

Even the state-of-the-art tool VAMB [37] was used for contig binning, the number of NCMAGs from long-read (NCMAG: S1 = 6, S2 = 6; assembled by metaFlye) and short-read assemblies (NCMAG: S1 = 0, S2 = 0, assembled by metaSPAdes; S1 = 1, S2 = 0, assembled by MEGAHIT) remained substantially lower than linked-read assembly (NCMAG: S1 = 21, S2 = 21; assembled by Pangea).

4. Ln 441-443: "In our experiments, we found that long-read assemblies had 60.98% fewer NCMAGs than linked-read assemblies from Pangea (Supplementary Note 4), indicating that some microbes might be lost due to insufficient long-read sequencing depth."

I don't understand why PacBio and ONT on the Zymo and ATCC datasets are downsampled? They only represent 0.5-2% of Gbp compared to the linked short reads. Please compare to when using a realistic throughput of data from these technologies. Because of this, currently, the results sections comparing Pangea assembly statistics to long-read based data does not really make sense.

For instance, ATCC-MSA-1003 has been used for assembly and binning of PacBio reads in Benoit et al., where assembly and binning using metaMDBG generate circularized genomes and almost complete genomes (Benoit et al., biorxiv, 2023). Likewise, ONT data has been used to generate many NCMAGs from various datasets.

Response: Sorry for the confusion. In lines 441-443, we compared the performance of Pangea (linked-reads) with metaFlye (long-reads) on S1 and S2. We did not perform long-read downsampling for these two human gut metagenomic sequencing datasets (S1: 6.26G; S2: 8.39G), which is comparable with a realistic setting. We only performed long-read downsampling

on Zymo and ATCC-MSA-1003 only for investigating hybrid assembly. Because hybrid assembly aims to explore a cost-effective way to improve metagenome assembly using high depth of short-reads and shallow depth of long-reads. We agree sufficient long-reads could significantly improve assembly continuity and generate circularized genomes, but our aim for this study is to explore a cost effective way for metagenome sequencing. We revised the manuscript as

Line 373-377:

As long-read sequencing has been limited in the applications in large-scale cohorts due to its high cost [12], the hybrid assembly combining short-reads (high depth) and long-reads (typically with shallow depth) in assembly was proposed as a cost-effective way to produce high-quality assemblies [15]. We evaluated the generalizability of Pangea on hybrid assembly by attaching virtual barcodes to short-reads using shallow depth long-reads (Methods).

5. Costs is mentioned several times as an advantage of the barcoded reads compared to long read, however no information on costs is given. What are the costs of generating the data linked data compared to short-read and long-read?

Response: Thanks for the comments. We compared the sequencing cost of TELL-Seq (~100G), short-read (~100G), HiFi (~100G), ONT (~100G) in the table below. The costs were estimated based on our most recent sequencing experiences. We found the price of long-read sequencing was almost 10 times compared to linked-read and short-read sequencing for the same volume of data.

Sequencing	Sequencing Library Preparation	Sequencing (~100Gb)	TELL-Seq™ Microbial Library Reagent Box 1 V1, RUO	Total
TELL-Seq	44.74 USD	383.49 USD	95.83 USD	524.06 USD
Short-read (Illumina)	44.74 USD	383.49 USD	0 USD	428.23 USD
HiFi	227.9 USD	6391.91 USD	0 USD	6619.81 USD
ONT	227.9 USD	3890.73 USD	0 USD	4118.63 USD

6. First paragraph in the Results section “Workflow of Pangea” is a bit difficult to follow. This makes it difficult to intuitively understand how the method works. Please consider re-writing.

Response: Sorry for any confusion caused by the unclear description. We have revised the first paragraph to make it easy to be followed:

Lines 121-140:

(i) Co-barcoded read binning. This module is intended to reduce the complexity of metagenomic sequencing data and is mainly used to improve the assemblies of high- and medium-abundance microbes. Pangaea extracts k-mer histograms and tetra-nucleotide frequencies (TNFs; Methods) of co-barcoded reads and represents them in low-dimensional latent space by Variational Autoencoder (VAE; Methods; Supplementary Note 2). Pangaea adopts a weighted sampling strategy on training VAE to balance the number of co-barcoded short-reads from microbes with different abundances (Methods). Pangaea utilizes RPH-kmeans (k-means based on random projection hashing) [31] to group co-barcoded short-reads in the latent space [31], which is beneficial for bins with uneven sizes (Methods). Short-reads from the same bin have a high chance of originating from the same microbe. These short-reads are then independently assembled. (ii) Multi-thresholding reassembly. Read binning may divide co-barcoded short-reads from the same low-abundance microbe into different bins. It could lead to poor assembly performance for these microbes due to insufficient data. Pangaea improves the assemblies of low-abundance microbes by collecting and reassembling the linked-reads that cannot be aligned to high-depth contigs obtained from read binning. The high-depth contigs are defined based on different depth thresholds (Figure 1 b; Methods). (iii) Ensemble assembly. This module is to eliminate the impact of mis-binning on final assembly results. Pangaea merged the assemblies from the previous two modules, the local assembly of Athena and the original short-read assembly using OLC assembly strategy (Methods). For short-reads with virtual barcodes, Pangaea would additionally integrate the contigs from the selected hybrid assembler using quickmerge (Methods).

7. Ln 190-199: I am not convinced that Thresholding improves the binning. There is no statistical test done and inspecting Supplementary Table 5 is a mix of blue and red (ie. better/worse). For many the differences are also very small in the number of bp. Maybe a statistical test could show whether it is better.

Response: Because of the numbers of low-abundance species (<1%) are quite small in ZYMO (8 species) and ATCC-MSA-1003 (10 species) for a statistical test, we simulated linked-reads from a complex community (CAMI-high) with 596 genomes. All of the microbes in CAMI-high with abundance below 1%. We found that the assembly with the thresholding had significantly higher genome fractions than those without thresholding (Wilcoxon paired rank-sum test p-value = 1.99e-05). We revised the manuscript as:

Line 191-201:

To demonstrate the performance of multi-thresholding reassembly, we compared the assemblies with and without multi-thresholding reassembly on both the TELL-Seq dataset of ATCC-MSA-1003 (low-complexity), and the stLFR dataset of CAMI-high (high-complexity). In the evaluation, we only consider the strains with abundances < 1% as low-abundance microbes. Specifically, we identified 10 such strains in the ATCC-MSA-1003 dataset and 596 strains in the CAMI-high dataset. On ATCC-MSA-1003, we found this module could increase the NGA50s of 5 low-abundance microbes (out of 6 strains, covered by more than 50% genomes; Supplementary Table 5). It could generate more sequences from the long contigs (>10Kb) of 6 low-abundance microbes (out of 7 strains with contigs longer than 10Kb; Supplementary Table 5). The assembly with multi-thresholding reassembly generated significantly higher genome fractions for the 596 low-

abundance microbes than the assembly without this module (Wilcoxon paired rank-sum test p-value = 1.99e-05; Supplementary Table 6).

8. Ln 162-163: “In order to enable these long-read binning tools on linked-reads, the co-barcoded reads were connected using a single “N” into long-reads”. For the comparison to long read binners, was the order of the linked reads known? Connecting the co-barcoded short reads with N in a random fashion is likely going to be difficult for the binners.

Response: The orders of the co-barcoded linked-reads are unknown. As the long-read binners will skip k-mers containing N (see code in METABCC-LR; the same for LRBinner), the process of connecting co-barcoded reads using “N”s will not affect the performance of long-read binners. This part is not directly related to metagenome assembly, so we moved this part to Supplementary Note 2.

Minor points:

9. Please reference Nissen et al., as they are the first to introduce VAEs for binning which is used here (disclaimer: I am a co-author of that paper).

Response: Thank you for the comments. We have cited the reference Nissen et al in our revised manuscript:

Line 527-529:

The normalized k-mer histogram (X_A) and TNF vector (X_T) were concatenated into a vector with 536 dimensions as the input to a VAE (Figure 1 b; Supplementary Note 2; the use of VAE was inspired by Nissen et al. [37]).

10. Ln 134-135: “We observed there was a negative correlation between $\max(X_A)$ and the abundance of the co-barcoded reads (Supplementary Figure 1).” I don’t understand why this is a negative correlation, but perhaps it is because I am not fully sure what $\max(X_A)$ is? Also I don’t understand how that can be seen from Supplementary Figure 1.

Response: Thank you for the comments. $\max(X_A)$ is the maximum value of the X_A vector. X_A is the histogram of k-mer frequencies, akin to the figure 7 in this previous study (Laehnemann, Borkhardt, and McHardy 2016) (copied below). According to this figure, the first peak predominantly consists of erroneous k-mers, and the second peak corresponds to the k-mers present in the reads.

Figure 7 in (Laehnemann, Borkhardt, and McHardy 2016). The first peak of the distribution is formed by very low coverage error k-mers and is usually modelled by a Poisson or a Gamma distribution. The second peak results from the majority of correct k-mers and is usually modelled by a Poisson or a Gaussian distribution. Between these two peaks, a clear local minimum can provide a k-mer trust coverage cut-off. The heavy tail of higher multiplicity k-mers is the result of k-mers from sequence repeats.

Another paper (Wickramarachchi et al. 2020) also used this sequence feature in long-read binning. Figure 2 (a) in this paper (copied below) describes X_A . From this figure, we can see that when the abundance becomes smaller (from high, medium, to low-abundance), the second peak of true k-mers (reflecting the abundance) gradually moves towards the first peak of the erroneous k-mers. Because the area below the curve is always 1 (this is a histogram), this moving of the true k-mers will result in a larger peak value of the entire curve (i.e., $\max(X_A)$). Actually, the two peaks become a single peak for the low-abundance band in this figure.

Figure 2 in (Wickramarachchi et al. 2020). The k-mer histogram of reads from species of three different abundance bands.

Supplementary Figure 5 (copied below) in our manuscript is the k-mer histogram of the linked-reads of four levels of abundances. At an abundance of 0.02%, the highest X_A value is 0.77. For abundance levels of 0.18%, 1.8%, and 18%, the corresponding X_A values are 0.44, 0.09, and <0.05 , respectively. This observation shows that there is a negative correlation between $\max(X_A)$ and the abundance of the co-barcoded reads.

Supplementary Figure 5. The k-mer histograms of co-barcoded reads of different abundance levels on the stLFR linked-reads of ATCC-MSA-1003. The line for abundance 18% is truncated at the global 15-mer frequency at 100 for better visualization of the other three abundance levels.

We have revised the section “Weighted sampling for VAE training” in Methods to interpret the negative correlation:

Line 550-560:

We designed weighted sampling to balance the training set from microbes with different abundances. The k-mer histogram (X_A) is the combination of two Poisson distributions; one represents the erroneous k-mers (always with a low frequency) [47], and the other represents the true k-mer abundances [43]. If the co-barcoded reads of a specific barcode are from a low-abundance microbe, the Poisson distribution of the true k-mer abundance would also have its peak value obtained at low frequency. Therefore, the peak of the two Poisson distributions will be stacked together and result in a larger $\max(X_A)$ (validated in Supplementary Figure 5). Considering that the highest value of X_A is negatively related to the abundance of the co-barcoded reads, we used a heuristic function $\max(X_A)^2$ as the sampling weight for the barcode of the co-barcoded reads. The square was to make the low-abundance microbes have a much higher sampling weight. The calculated sampling weights were automatically used by the `WeightedRandomSampler` of PyTorch to construct a balanced training dataset.

11. Ln 137: Describe RPH-kmeans, what is RPH?

Response: RPH stands for random-projection hashing. This algorithm was proposed in a previous paper (Xie et al. 2020). We have revised Line 127-129 of Results to interpret PRH:

Line 127-129:

Pangaea utilizes RPH-kmeans (k-means based on random projection hashing) [31] to group co-barcoded short-reads in the latent space [31], which is beneficial for bins with uneven sizes (Methods).

12. Ln 141-142: “This module aligns short-reads to contigs and removes them if they are from the contigs with abundances above certain thresholds”. Please elaborate a bit more.

Response: Thank you for the comment. We revised this sentence as below:

Line 131-136:

(ii) Multi-thresholding reassembly. Read binning may divide co-barcoded short-reads from the same low-abundance microbe into different bins. It could lead to poor assembly performance for these microbes due to insufficient data. Pangaea improves the assemblies of low-abundance microbes by collecting and reassembling the linked-reads that cannot be aligned to high-depth contigs obtained from read binning. The high-depth contigs are defined based on different depth thresholds (Figure 1 b; Methods).

The detailed process of this module is described in Methods:

Line 565-571:

To calculate the read depth of the contigs assembled from each read cluster (denoted as $\text{contigs}_{\text{bin}}$), we aligned the input reads to $\text{contigs}_{\text{bin}}$ using BWA-MEM (v0.7.17) [42] and calculated the read depth for each contig using the “jgi summarize bam contig depths” program in MetaBat2 (v2.12.1) [48]. To collect the reads of low-abundance microbes, we extracted the reads that could

not be mapped to the high-depth contigs (with read depth > ti) in contigs_{bin} and assembled them using the standard short-read assembler, metaSPAdes (v3.15.3) [29].

13. Ln 171-187: Comparison between ASM_b and ASM_notB. Can you report precision/recall instead of N50 and NGA50? Ie what was the precision/recall of a genome reconstruction between the two approaches?

Response: We performed contig binning for CAMI, S1, S2, and S3 to report contamination precision (1 - precision) and completeness (recall) of MAGs (Supplementary Table 4, and 12). In the main text, we used NCMAG to represent the MAGs with high completeness and low contamination.

Line 165-173:

We further evaluated the assemblies with respect to MAG qualities (completeness, contamination, and RNA annotations) after contig binning on CAMI-high and the three human gut microbiomes (Methods). For ATCC-MSA-1003, the reference genomes are available, and the community design is simple (only 20 strains); therefore, contig binning is unnecessary. We observed ASM_B generated one more NCMAG than ASM_{-B} for CAMI-high (ASM_B = 14, ASM_{-B} = 13; Supplementary Table 4). For the human gut microbiomes, ASM_B obtained more NCMAGs than ASM_{-B} on all the three samples (S1: ASM_B = 24, ASM_{-B} = 20; S2: ASM_B = 17, ASM_{-B} = 11; S3: ASM_B = 9, ASM_{-B} = 8; Supplementary Table 4). These NCMAGs from ASM_B were commonly observed from high-abundance microbes (average depths of NCMAGs: S1 = 526.6X; S2 = 211.19X; S3 = 256.52X).

14. Ln 180: What is bin number, I assume this is number of clusters? This has not been introduced before. Do you have to set a number beforehand? How is this number set?

Response: The bin number refers to the number of clusters for co-barcoded reads binning. It is a hyperparameter of Pangaea, which is positively related to the Shannon diversity of the dataset ($\alpha \times \text{Shannon diversity}$). It is a trade-off for bin number selection: choosing a low bin number might result in limited power to deconvolve dataset complexity, and a high bin number might lead to poor assembly due to separating reads of the same microbe into different bins. We manually set 30 for the three human gut microbiomes (they have close Shannon diversity), which could generate the best assembly. This enables us to calculate $\alpha=8$ and bin numbers can be calculated as $8 \times \text{Shannon diversity}$. We added a paragraph to describe the process of determining the bin numbers:

Line 174-183:

We observed the number of read bins (k) for read binning was positively correlated with the diversities of metagenome sequencing datasets ($k = \alpha \times \text{Shannon Diversity}$; Supplementary Table 2; Methods). To determine the coefficient a, we chose a k that worked well on all three human gut microbiomes (k = 30) and calculated the coefficient α as 8 by linear regression. Our results showed that k could influence both the precision and recall of read binning (a large k resulted in high binning precision and low recall; Supplementary Figure 2). The k is a trade-off between generating read bins with low complexities (large k) or keeping more reads from the same

microbes in the same bin (small k). Although k would influence read binning performance, the assembly results seemed robust if k was not shifted too much from the value calculated from the formula ($k = 15$) for ATCC-MSA-1003 (e.g., assemblies from $k = 10, 15, 20$ were comparable; Supplementary Figure 3).

15. Figure 3.1. Please add meaningful label to y-axis (currently now is "value")

Response: We revised the label of the y-axis of Figure 3.1 as Value of ARI/F1/Precision/Recall.

References

- Laehnemann, David, Arndt Borkhardt, and Alice Carolyn McHardy. 2016. "Denoising DNA deep sequencing data—high-throughput sequencing errors and their correction." *Briefings in bioinformatics* 17 (1): 154-179.
- Van der Walt, Andries Johannes, Marc Warwick Van Goethem, Jean-Baptiste Ramond, Thulani Peter Makhwanyane, Oleg Reva, and Don Arthur Cowan. 2017. "Assembling metagenomes, one community at a time." *BMC genomics* 18 (1): 1-13.
- Vollmers, John, Sandra Wiegand, and Anne-Kristin Kaster. 2017. "Comparing and evaluating metagenome assembly tools from a microbiologist's perspective-not only size matters!" *PLoS one* 12 (1): e0169662.
- Wickramarachchi, A., V. Mallawaarachchi, V. Rajan, and Y. Lin. 2020. "MetaBCC-LR: metagenomics binning by coverage and composition for long reads." *Bioinformatics* 36 (Suppl_1): i3-i11. <https://doi.org/10.1093/bioinformatics/btaa441>. <https://www.ncbi.nlm.nih.gov/pubmed/32657364>.
- Xie, Kaikun, Yu Huang, Feng Zeng, Zehua Liu, and Ting Chen. 2020. "scAIDE: clustering of large-scale single-cell RNA-seq data reveals putative and rare cell types." *NAR genomics and bioinformatics* 2 (4): lqaa082.
- Zhang, Zhenmiao, Chao Yang, Werner Pieter Veldsman, Xiaodong Fang, and Lu Zhang. 2023. "Benchmarking genome assembly methods on metagenomic sequencing data." *Briefings in Bioinformatics* 24 (2): bbad087.

REVIEWER COMMENTS

Reviewer #1 (Remarks to the Author):

The authors have made substantial revisions based on the feedback from all reviewers. After the revisions, the quality of the manuscript has significantly improved. For instance, the authors have added a considerable amount of simulated sequencing datasets, incorporated a more comprehensive set of metrics, and compared their tool against a wider range of other tools for benchmarking (pertaining to Questions 1 to 4, 10, and 12).

At present, our only concern revolves around the choice of bin number and T (Questions 7 and 8), which are hyperparameters of Pangaea. The method remains unclear to us. We still wonder how these parameters should be determined, especially when users aim to apply this software to their own complex samples. This clarification is crucial, as we know that complex samples (those whose diversity or composition often differs from typical human microbiome samples) frequently require more intricate assembly processes. Is there a guideline available to help users specify the bin number and T? Can the default parameter be used for those complex samples as well? The authors should clarify this.

Reviewer #2 (Remarks to the Author):

No additional comments

Response to Reviewer Comments

Reviewer #1 (Remarks to the Author):

The authors have made substantial revisions based on the feedback from all reviewers. After the revisions, the quality of the manuscript has significantly improved. For instance, the authors have added a considerable amount of simulated sequencing datasets, incorporated a more comprehensive set of metrics, and compared their tool against a wider range of other tools for benchmarking (pertaining to Questions 1 to 4, 10, and 12).

At present, our only concern revolves around the choice of bin number and T (Questions 7 and 8), which are hyperparameters of Pangaea. The method remains unclear to us. We still wonder how these parameters should be determined, especially when users aim to apply this software to their own complex samples. This clarification is crucial, as we know that complex samples (those whose diversity or composition often differs from typical human microbiome samples) frequently require more intricate assembly processes. Is there a guideline available to help users specify the bin number and T ? Can the default parameter be used for those complex samples as well? The authors should clarify this.

Response: Thank you for your comments. Pangaea requires two hyperparameters, k (bin number) and T , which could be determined based on the following facts:

1. Hyperparameter k (bin number)

The k is a trade-off between generating read bins with low complexities (large k) or keeping more reads from the same microbes in the same bin (small k), and k should be positively correlated with the biodiversity of a metagenomic sample. In Pangaea, we used Shannon diversity to measure the biodiversity in a sample and calculated k as $a * \text{Shannon diversity}$. The weight a to adjust the calculation is 8 by default, which was calculated based on the real samples in our experiments.

We argue this setting is applicable to high-complex datasets for two reasons:

1) Based on our results, this setting obtained good assembly results for both low-complexity (TELL-Seq sequencing data of ATCC-MSA-1003; Shannon diversity = 1.89; $k = \text{round}(8 * 1.89) = 15$; lines 155-173 in the manuscript) and high-complexity metagenomic sequencing datasets (simulated stLFR sequencing data of CAMI-high; Shannon diversity = 4.57; $k = \text{round}(8 * 4.57) = 37$; lines 155-173 in the manuscript).

2) We performed two experiments:

1. Compare the assembly results of ATCC-MSA-1003 by varying k from 10 to 15 for ATCC-MSA-1003 (**Supplementary Figure 3**) and 2. Compare the assembly results of CAMI-high by varying k from 25 to 45 (**Supplementary Table 5**). We observed the assembly results were not sensitive to the setting of k for both ATCC-MSA-1003 and CAMI-high if k values were not too far away from $8 * \text{Shannon diversity}$.

Supplementary Figure 3. The assembly statistics (a), N_x (b), and N_{Ax} (c) of Pangaea assemblies on the stLFR linked-reads of ATCC-MSA-1003 based on different cluster numbers ($k=10, 15, \text{ and } 20$).

CAMI high	k=25	k=30	k=35	k=40	k=45
Total assembly length	798,567,819	798,074,124	799,834,811	798,362,051	798,962,833
Genome fraction (%)	27.74	27.70	27.72	27.72	27.78
Largest alignment	3,405,236	3,405,180	3,405,152	3,405,134	3,405,134

Overall N50	253,171	253,173	254,141	248,039	253,098
Overall N70	124476	123039	124320	123558	124503
Overall N90	38,577	38,467	38,713	38,665	38,692
Overall NA50	213,149	212,624	213,549	212,174	212,427
NA50 per strain	68,162	69,008	66,544	67,457	67,273
NGA50 per strain	55,298	55,200	54,265	55,475	55,206
Non-zero NGA50	197	197	194	196	196

Supplementary Table 5. The assembly statistics of Pangaea using different parameters k on CAMI-high.

We revised the last paragraph of section “Co-barcoded read binning improves assemblies for microbes with high and medium abundances” to interpret our strategy to determine k :

Lines 174-183:

Our results showed that the number of read bins (k) could influence both the precision and recall of read binning (a large k resulted in high binning precision and low recall; Supplementary Figure 2). The k is a trade-off between generating read bins with low complexities (large k) or keeping more reads from the same microbes in the same bin (small k). k for read binning was set linear to the biodiversity of a metagenomic sample ($k = a * \text{Shannon Diversity}$; Supplementary Table 2; Methods). To determine the coefficient α , we chose the k ($k=30$) that worked well on all three real metagenomic datasets and calculated the coefficient a as 8 by linear regression. This setting of k is applicable to both low- (ATCC-MSA-1003) and high-complexity (CAMI-high) datasets, and the assembly results seemed robust if k was not shifted too much from the value calculated from the formula (15 for ATCC-MSA-1003, **Supplementary Figure 3**; 37 for CAMI-high, **Supplementary Table 5**).

2. Hyperparameter T (read depth threshold)

T is set for reassembling the contigs for low-abundance microbes and it is not related to the biodiversity level of a sample. In Pangaea, we use threshold values of $T=10$ and $T=30$ to improve the assembly results for those microbial genomes that have a read depth below 10 and 30, respectively. In practice, $T=10$ and $T=30$ is applicable to the datasets with both low and high complexity. For the TELL-Seq sequencing data of ATCC-MSA-1003, the two thresholds on T improved the total assembly length (with T: 61.99Mb; without T: 60.08Mb; **Supplementary Table 6**) and genome fraction (with T: 82.63%; without T: 80.38%; **Supplementary Table 6**). These thresholds could increase the NGA50s of 5 low-abundance (defined as microbes of abundance $< 1\%$) microbes (out of 6 strains with non-zero NGA50s; **Supplementary Table 6**). The two thresholds could generate more sequences from the long contigs ($>10\text{Kb}$) of 6 low-abundance microbes (out of 7 strains with contigs longer than 10Kb; **Supplementary Table 6**). For the simulated stLFR sequencing data of CAMI-high, the two thresholds enable significant

improvement of genome fractions compared to the assembly without low-abundance reassembly ($p=1.99e-5$; **Supplementary Table 7**).

REVIEWERS' COMMENTS

Reviewer #1 (Remarks to the Author):

The authors have fully addressed our previous concern on the choice of two hyperparameters (k and T) in their method. We recommend the publication of this work.